JCB Journal of Cell Biology

**TOOLS**

# ExTrack characterizes transition kinetics and diffusion in noisy single-particle tracks

François Simon[1,2] , Jean-Yves Tinevez[3] , and Sven van Teeffelen[1,2]

Single-particle tracking microscopy is a powerful technique to investigate how proteins dynamically interact with their environment in live cells. However, the analysis of tracks is confounded by noisy molecule localization, short tracks, and rapid transitions between different motion states, notably between immobile and diffusive states. Here, we propose a probabilistic method termed ExTrack that uses the full spatio-temporal information of tracks to extract global model parameters, to calculate state probabilities at every time point, to reveal distributions of state durations, and to refine the positions of bound molecules. ExTrack works for a wide range of diffusion coefficients and transition rates, even if experimental data deviate from model assumptions. We demonstrate its capacity by applying it to slowly diffusing and rapidly transitioning bacterial envelope proteins. ExTrack greatly increases the regime of computationally analyzable noisy single-particle tracks. The ExTrack package is available in ImageJ and Python.

## Introduction

Studying the motion of proteins by single-particle tracking (SPT) allows to characterize how proteins dynamically interact with their environment (Manley et al., 2008; Kusumi et al., 2014). Notably, SPT can reveal if and where proteins are diffusive or immobile (Persson et al., 2013; Uphoff, 2016; Martens et al., 2019). This information has significantly improved our understanding of important biological processes such as transcription-factor binding dynamics, antibody recognition, cytoskeletal dynamics, or intracellular transport (Kusumi et al., 2014; Pierobon et al., 2009; Monnier et al., 2015; Stracy et al., 2016; Jézéquel et al., 2018; Callegari et al., 2019; Li and Xie, 2011; Izeddin et al., 2014; Özbaykal et al., 2020).

Molecules often transition between different motion states. If transitions happen rarely and if trajectories are long, different states such as immobile or diffusive states are reliably detected from time-averaged quantities such as the mean-squared displacement (Michalet, 2010; Elliott et al., 2011; Bosch et al., 2014; Hansen et al., 2018). However, molecules often undergo rapid transitions between different states (Martens et al., 2019; Callegari et al., 2019; Pierobon et al., 2009). Furthermore, tracks are often short as particles can bleach or diffuse out of the field of view or focal plane (Hansen et al., 2018). In such situations, probabilistic methods are better suited to determine global parameters such as diffusion coefficients and transition rates (Dempster et al., 1977; Persson et al., 2013; Meent et al., 2013; Calderon, 2014; Slator et al., 2015; Monnier et al., 2015; Smith

et al., 2019; Falcao and Coombs, 2020; Karslake et al., 2021; Vink et al., 2020a; Rahm et al., 2021). Some of these methods can also predict the motion states of individual molecules at every time point (Persson et al., 2013; Monnier et al., 2015; Briane et al., 2020; Rahm et al., 2021), which can reveal the locations of binding sites, spatial correlations, and complex, potentially non-Markovian dynamics (Mahmutovic et al., 2012).

Previous probabilistic methods for diffusive models shown to correctly estimate diffusion and transition parameters (Persson et al., 2013; Vink et al., 2020a) are based on absolute distances between subsequent localizations. These methods have been developed for situations where physical displacements are large in comparison with the localization uncertainty for each molecule. However, when molecules transition rapidly between states, high time resolution is needed, which results in small physical displacements, which, in turn, make identifying different motion states hard or impossible (Fig. 1, a–c). On the contrary, the whole track still allows the distinction of states (Fig. 1 a), simply because subsequent positions of immobile or slowly diffusing molecules fall in the same small area determined by localization error, while subsequent positions of fast-diffusing molecules are nearly uncorrelated.

To account for those spatial correlations, the full sequence of track positions must be taken into account. This approach has been used to characterize a single population of diffusing

[1]Département de Microbiologie, Infectiologie, et Immunologie, Faculté de Médecine, Université de Montréal, Montréal, Quebec, Canada;   [2]Microbial Morphogenesis and Growth Lab, Institut Pasteur, Université de Paris Cité, Paris, France;   [3]Image Analysis Hub, Institut Pasteur, Université de Paris Cité, Paris, France.

Correspondence to Sven van Teeffelen: sven.vanteeffelen@gmail.com.

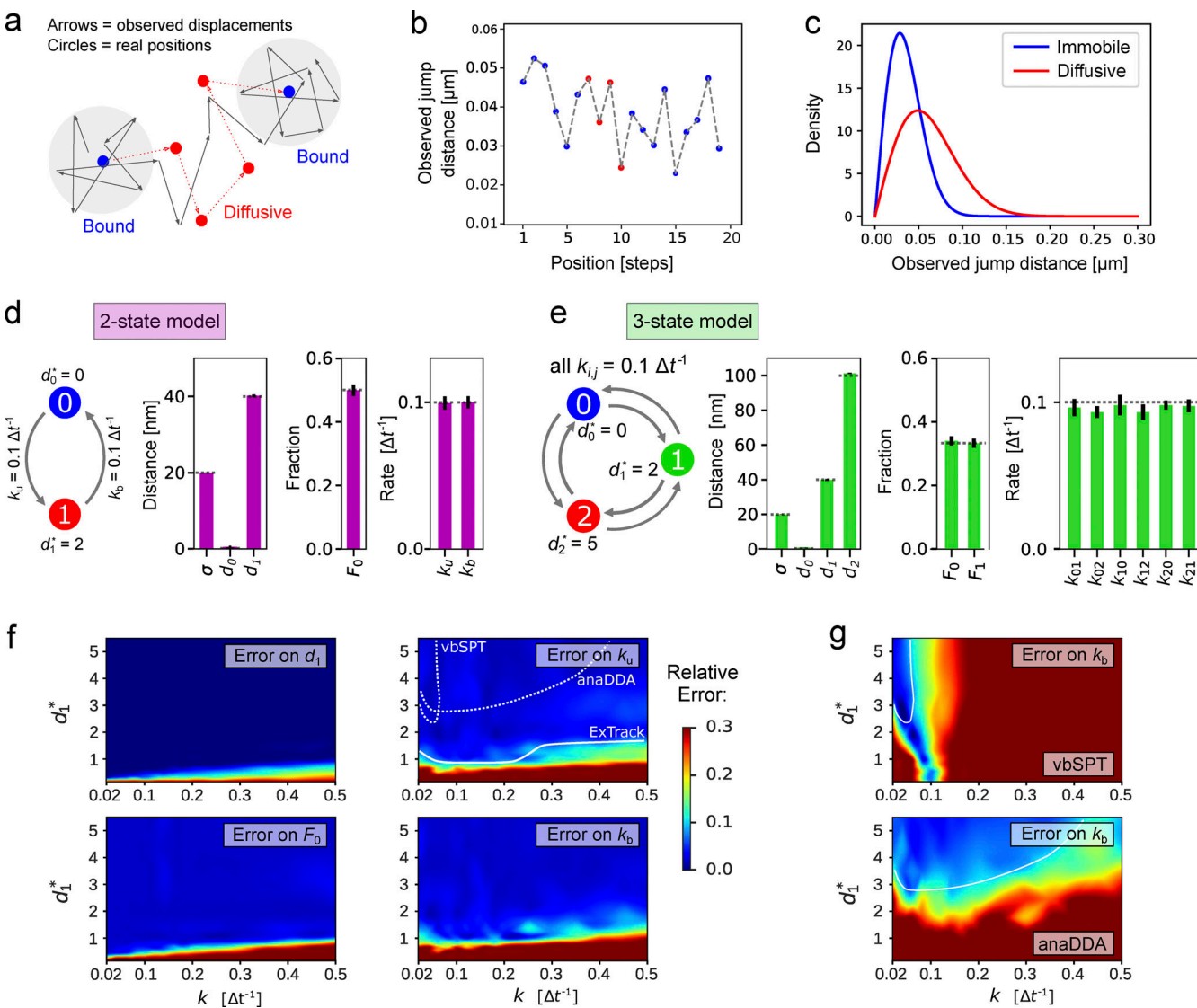

Figure 1. **ExTrack permits to assess a wide range of multi-state diffusion models. (a)** Example track of a molecule transitioning between immobile and diffusive states with $d_1 = 2\,\sigma$. Arrows: observed displacements; dots: actual positions of immobile (blue) and diffusive (red) molecules. **(b)** Consecutive observed distances of the track from a. **(c)** Density function of observed distances of coefficiently immobile (blue) or diffusive (red) molecules for $d_1 = 2\,\sigma$. **(d and e)** Left: Simulated two-state (d) and three-state (e) diffusion models with diffusion length and transition rates as indicated. Right: Model parameters estimated by ExTrack (mean ± SD) assuming a two-state (d) or three-state (e) model (localization error $\sigma$, diffusion lengths $d_0$ and $d_1$, initial immobile fraction $F_0$, transition rates $k$). Dotted lines: ground truth. ExTrack settings: two-state data: two sub-steps, window length = 10; three-state data: no sub-steps, window length = 7. **(f)** Heatmap of the relative errors of $d_1$, $F_0$, $k_u$, and $k_b$ obtained from a two-state model fit to two-state simulations as in d. Error: mean absolute relative errors from 10 replicates per condition. White lines indicate regions of <10% error for model parameters hardest to fit for ExTrack ($k_u$, solid), vbSPT, and anaDDA ($k_b$, dashed, see g). Initial parameters: $d_0 = 0\,\mu m$, $d_1 = 0.1\,\mu m$, $\sigma = 0\,\mu m$, $k = 0.1\,\Delta t^{-1}$, $F_0 = 0.4$. **(g)** Error on $k_b$ of vbSPT and anaDDA (same protocol and color map as in f, 20 replicates were used for anaDDA to mitigate its variability). See Fig S2 a for errors on the other parameters.

molecules (Berglund, 2010; Relich et al., 2016). However, if molecules transition between states, this approach becomes computationally demanding because all possible sequences of single-molecule states need to be considered. To avoid this computational complexity, different mean-field approximations (Calderon, 2014; Slator et al., 2015; Bernstein and Fricks, 2016; Lindén et al., 2017; Lindén and Elf, 2018) and machine-learning approaches (Kowalek et al., 2019; Chen et al., 2021) have been proposed. However, their performance across model parameters remains to be investigated.

Here, we propose an alternative probabilistic method to extract diffusive motion states and transitions: We tackle the combinatorial problem of different motion states by introducing a sliding window approximation that maintains the most important spatio-temporal correlations. The method is fast and accurate for a large range of parameters, even if physical displacements are similar to the localization error. The method is also robust with respect to deviations between data and model assumptions. Additionally, the method annotates the state probabilities at the single-molecule level, refines localizations (Lindén and Elf, 2018), and extracts distributions of state

durations. We demonstrated its versatility by analyzing two bacterial membrane proteins that diffuse slowly and transition rapidly between immobile and diffusive states.

## Results

### ExTrack is a maximum-likelihood method to detect different diffusion states in single-molecule tracks

We developed ExTrack, a maximum-likelihood estimation (MLE) method that contains two main modules: A fitting module fits a multi-state Markovian diffusion model to a data set of noisy single-molecule tracks. This module infers global model parameters including localization error, diffusion lengths, transition rates, and the initial fractions of molecules (at the first time point of all tracks). Part of these global parameters can also be provided by the user, and localization error can even be provided for each peak (Thompson et al., 2002; Quan et al., 2010) if desired. ExTrack is flexible in terms of the number of states and spatial dimensions. Additionally, it can explicitly consider molecules leaving the field of view, which otherwise introduces bias (Hansen et al., 2018). Based on global parameters, a single-molecule annotation module then estimates state probabilities for molecules to reside in each state at each time point. To characterize single-molecule tracks further, we developed two additional modules: A position-refinement module refines molecule positions by taking advantage of spatial correlations between subsequent localizations, conceptually similar to Lindén and Elf (2018). This feature allows to maintain high temporal resolution for state transitions, while attaining high spatial resolution for immobile molecules. A fourth module produces histograms of state durations to reveal potential non-Markovian transitions.

ExTrack is based on a Hidden Markov Model (HMM) that approximates a continuous-time process by a discrete-time Markov model (Das et al., 2009). The method calculates the probability density of observing each track given a set of global parameters (Lindén et al., 2017). In principle, this calculation requires to integrate the joint probability density of states, real positions, and observed positions over all possible sequences of hidden diffusive states and over all sequences of hidden real (physical) molecule positions (see Materials and methods, ExTrack fitting module). This integration can be performed analytically by computing the sequences $\mu$ and $s$ (Eq. 6, Materials and methods). However, this problem is computationally intractable for long tracks: Indeed, the computation time scales with the number of possible sequences of states which increase exponentially with the number of time points. To reduce computation time, we took advantage of the fact that the real position at a given time step is little influenced by the actual state several time points away. This allows us to introduce a sliding-window approximation that greatly limits the computation time of ExTrack. More specifically, when recursively computing the probability density of observing a track, we need to consider all possible combinations of states up to the current time point of the integration. To save computational time, the window method averages over different sequences of states outside of the sliding window, while explicitly considering

different combinations of states within the window (see Materials and methods, Using a window…). Choosing the window length is a compromise between accuracy and speed. We suggest using a window length of 3 to 7 depending on the expected diffusion lengths. We will show later that a window length of 3 can be sufficiently large if the ratio between diffusion length and localization precision is greater than about 2.

In many HMMs, it is assumed that state transitions can only occur at the time points of the measurement (Das et al., 2009). However, this approximation introduces a bias toward higher fractions of fast diffusing molecules. Instead, we assume state transitions to occur at the middle of steps (see Materials and methods, Parameter fitting…). Additionally, ExTrack can consider sub-steps to further reduce bias at high transition rates.

ExTrack is available both as a Python library (Simon, 2022) and as a TrackMate module (Ershov et al., 2022) on Fiji.

### Performance and comparison to alternative methods

First, we tested the performance of the ExTrack fitting module by applying it to computationally simulated noisy tracks of molecules (10,000 tracks of 10 positions each, if not stated otherwise) that transition between an immobile state (state 0) and a slowly diffusive state (state 1). The latter is characterized by a small diffusion length $d_1 = 2\,\sigma$, where $\sigma$ is the localization error (Fig. 1 a). The diffusion length is the typical physical displacement: $d_1 = \sqrt{2D_1\Delta t}$ with $D_1$ the diffusion coefficient and $\Delta t$ the time step. Here, we assume symmetric binding and unbinding rates $k_u = k_b = 0.1\,\Delta t^{-1}$. Thus, on average, molecules reside in each state for 10 time steps.

The dimensionless parameter $d_1^* = d_1/\sigma$ can be regarded as a signal-to-noise ratio. For a typical experiment, with $\sigma = 20$ nm and a time step of $\Delta t = 20$ ms, a rescaled diffusion length of $d^* = 2$ corresponds to a diffusion coefficient of $D_1 = 0.04\ \mu m^2\ s^{-1}$, which is representative of typical membrane proteins in vivo (Kumar et al., 2010; Oswald et al., 2016).

ExTrack reliably estimates all global model parameters (Fig. 1 d) despite similar observed distances for immobile and diffusive molecules (Fig. 1 c) and despite a low number of 10 localizations per track. These estimates are robust with respect to the initial parameters (for example, if varying the initial $d^*$ between 0.3 and 10). Since molecules are considered at steady state in our example, the initial fraction of immobile molecules is given by $F_0 = \left(\frac{k_b}{k_u + k_b}\right)$.

Next, we simulated tracks for a three-state model, with $d_0^* = 0$, $d_1^* = 2$, and $d_2^* = 5$, where transition rates are $k_{i,j} = 0.1\,\Delta t^{-1}$ for all pairs of states. Fig. 1 e demonstrates that ExTrack estimates global model parameters reliably. If the data contain long enough tracks, ExTrack can also correctly predict two immobile states of different lifetimes and their associated transition rates (Table S1). We will revisit more complex data sets below.

Returning to the simpler two-state model, ExTrack is capable of predicting global model parameters reliably for a large range of model parameters (Fig. 1 f). Predictions are accurate for diffusion lengths as low as the localization error and transition rates as high as 0.5 $\Delta t^{-1}$ (for independent variations of $k_b$ and $k_u$, see Fig. S1 a). To account for rapid transitions and low diffusion

lengths, we employed ExTrack considering two sub-steps and a window length of 10. However, the method predicts parameters almost equally reliably without sub-steps (Fig. S1), or with a smaller window size of 3 (Fig. S1 c) while achieving improved computation time (Fig. S1 d).

Next, we compared ExTrack with the two MLE-based methods, vbSPT (Persson et al., 2013) and anaDDA (Vink et al., 2020b) that use absolute distances between localizations for parameter estimation. While vbSPT uses an HMM for the likelihood estimate (Persson et al., 2013), anaDDA is based on an analytical form of the distributions of apparent diffusion coefficients from short tracks (Vink et al., 2020b). Both methods are restricted to a smaller parameter range than ExTrack (Fig. 1, f and g; and Fig. S2 a) in the tested regime. The errors of parameter estimation by vbSPT are largely due to systematic bias, while the error of anaDDA is predominantly stochastic (Fig. S2 d; Vink et al., 2020b). We also tested a mean-field approximation based on track positions and considering hidden particle positions, the variational method UncertainSPT (Lindén et al., 2017). We found that UncertainSPT performs worse and takes more computation time than ExTrack, anaDDA, or vbSPT (Fig. S2 b).

### ExTrack is robust with respect to non-ideal motion properties
Single-molecule tracks in real cells often deviate from our basic model assumptions. Here, we investigated three different types of such deviations: (i) variations of diffusion coefficients or localization precision, (ii) finite track lengths due to a finite field of view or focal depth, and (iii) physical confinement.

Diffusion coefficients can show intra- or inter-track variations (Stylianidou et al., 2014; Slator et al., 2015; El Beheiry et al., 2015), for example due to local variations of viscosity (Stylianidou et al., 2014), and localization error can vary, for example if molecules are out of focus. We thus simulated tracks of a two-state model with diffusion coefficients or localization precision drawn from a chi-squared distribution with fixed mean and variable coefficient of variation. First, we show that ExTrack gives very accurate predictions when localization error is specified for each peak instead of being treated as a single global fitting parameter (Fig. S3 a). However, even when no prior information on localization error is given, ExTrack reliably predicts the average model parameters for variations up to 30–50% (Fig. 2 a), in contrast to the distance-based methods anaDDA and vbSPT (Fig. S3 c). Track-to-track variations in diffusion coefficient of similar magnitude (up to about 50%) also do not affect predictions of average parameters (Fig. 2 b).

In situations, where the diffusion coefficient is even more broadly distributed, ExTrack can be used assuming a three-state model followed by aggregation of two diffusive states (Fig. S3 b). We tested this aggregation approach with simulations of one immobile and five diffusive states, mimicking a broad distribution of diffusion lengths and jump distances (Fig. S3 b and Table S2). The aggregated three-state approach reliably quantifies transitions between aggregated states and corresponding state fractions, thus providing a practical approach to the often-encountered difficulty of choosing the right number of diffusive states.

Second, molecules can leave the field of view depending on microscopy modality and substrate geometry. For example,

cytoplasmic molecules studied by confocal or epi-fluorescence microscopy diffuse in and out of the focal plane, and proteins embedded or attached to a cylindrical membrane (for example, in bacteria) studied by Total Internal Reflection Fluorescence (TIRF) microscopy leave the illumination field (Fig. 2, c and d). Thus, immobile or slowly diffusing molecules are over-represented among long tracks, which have previously been described as "defocalization bias" (Hansen et al., 2018). We alleviate this bias by taking track termination into account explicitly (Materials and methods, Extension of ExTrack...) similarly to previous approaches (Kues and Kubitscheck, 2002; Hansen et al., 2018). In both free 3D diffusion and diffusion along a cylindrical membrane, ExTrack reliably estimates model parameters as long as the typical dimension (focal depth or width of the field of view) is at least twice the diffusion length (Fig. 2, c and d).

Finally, we tested the ability of ExTrack to analyze tracks of spatially confined molecules, as frequently found in membrane domains or small volumes such as bacteria or intracellular compartments. ExTrack performed robustly as long as confining dimensions are at least two to four times larger than the diffusion length (Fig. 2 e). Interestingly, while the diffusion length is underestimated for confining dimensions smaller than about 4 $d$, the transition rates are predicted reliably even if the confining box size is as small as 2 $d$.

### ExTrack computes state probabilities at every time point and refines positions
Next, we tested the performance of the single-molecule probabilistic annotation module of ExTrack, which is based on global model parameters and annotates state probabilities for every time point (Bernstein and Fricks, 2016).

Fig. 3, a and b, shows tracks from the simulation of a two-state model with an immobile and a slowly diffusive state ($d^* = 2$). Despite the small value of $d^*$, motion states are reliably estimated in this example. Then, we estimated the predictive power of the probabilistic annotation module by applying it to large sets of tracks with different values of $d^*$ (0.5, 1, or 2). As expected, higher $d^*$ values result in higher confidence in predicting states (Fig. 3 c). To demonstrate the accuracy of the probabilistic annotation, we confirmed that among all molecules predicted to reside in the diffusive state with probability $p_1$, the fraction of molecules actually diffusive also equals $p_1$ (Fig. S4 a).

Previous methods often classify molecules categorically into the most likely state (Forney, 1973; Persson et al., 2013; Lindén et al., 2017). We used this approach to measure the performance of ExTrack and compare it to previous methods: First, we estimated the accuracy of categorical state annotations depending on the diffusion length and the transition rates (Fig. S4 b). Increasing transition rates led to worse estimates because states are easier to estimate at time points that are distant from transitions (Fig. S4 c). Second, we found ExTrack annotations to be robust with respect to wrongly chosen global parameters (Fig. S4 d). Finally, we found that the ExTrack state annotation module performs better than vbSPT (Fig. S4 e).

Next, we tested the capacity of ExTrack to refine positions by calculating the most likely physical position for each time point. Fig. 3, d and e, demonstrates that the position-refinement

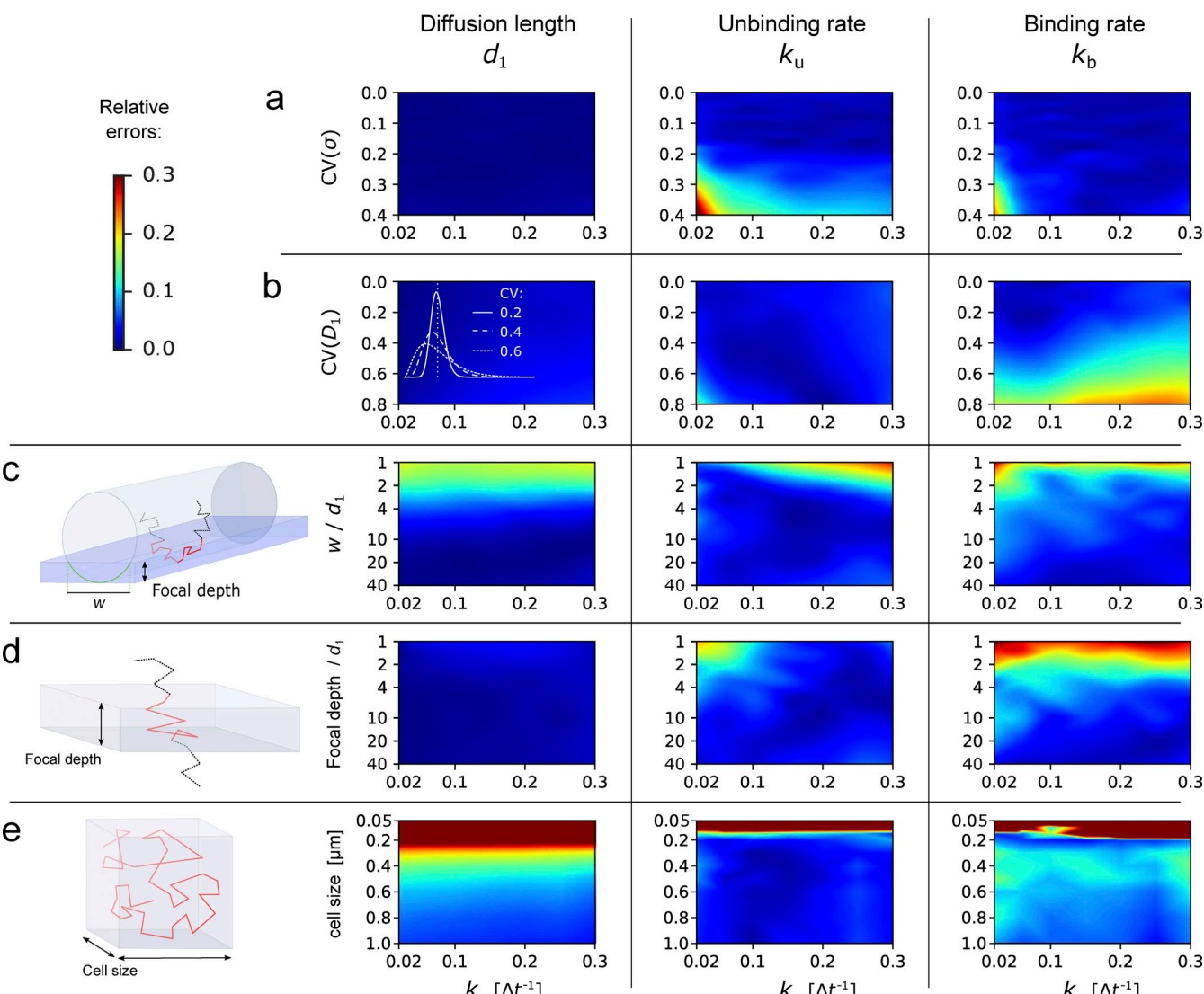

Figure 2. **Robustness of ExTrack to various sources of bias.** Heatmaps of relative errors on $d_1$, $k_u$, and $k_b$ in case of two-state parameter fits to two-state simulations with one immobile state and one diffusive state, for different sources of bias, as a function of the source of bias (y axis) and transition rates $k$. **(a and b)** We simulated track-to-track variations of localization error $\sigma$ (a) or diffusion coefficient $D_1$ (b). Varied parameters followed chi-square distributions (white graphs in b) re-scaled so the mean localization error equals 0.02 µm (a) or the mean diffusion coefficient equals 0.25 µm².s⁻¹, which corresponds to a diffusion length of $d_1^* = 5$ (b). **(c)** Membrane proteins diffuse on a cylindrical surface and leave the field of view on the sides (see cartoon). We varied the width $w$ of the field of view as indicated in the cartoon, while maintaining $d_1$ and $\sigma$ fixed. **(d)** Cytoplasmic proteins can leave the focal plane anywhere (see cartoon). We varied the focal depth while maintaining $d_1$ and $\sigma$ fixed. **(e)** Particles confined in a symmetric cube. We varied the box size while maintaining $d$ and $\sigma$ fixed. 10 replicates per condition. If not stated otherwise, $d_0 = 0$ µm, $d_1 = 0.1$ µm, and $\sigma = 0.02$ µm. ExTrack settings: window length = 7, no sub-steps. CV, coefficient of variation.

module effectively reduces the localization error of immobile molecules by $\sqrt{N}$, where $N$ is the number of localizations in the immobile segment. This feature allows to obtain accurate positions of molecular binding sites inside cells, while still resolving state transitions dynamics.

**ExTrack computes distributions of state durations to characterize transition kinetics beyond the Markov transitions assumption**

ExTrack provides a histogram module that generates probability distributions of state durations. Instead of considering only the

most likely set of states, ExTrack considers a large number of potential state vectors and weighs them with their corresponding probabilities. To test the histogram module, we first simulated a two-state model with Markovian transitions. The predicted diffusive and immobile state durations are distributed exponentially, as expected, and in agreement with the simulated data (Fig. 3 f). Therefore, any deviation from exponential decay can reveal more complex transition behavior: As an example, we simulated molecules that transition between two immobile states and one diffusive state (Fig. 3 g). The histogram of immobile state durations then accurately

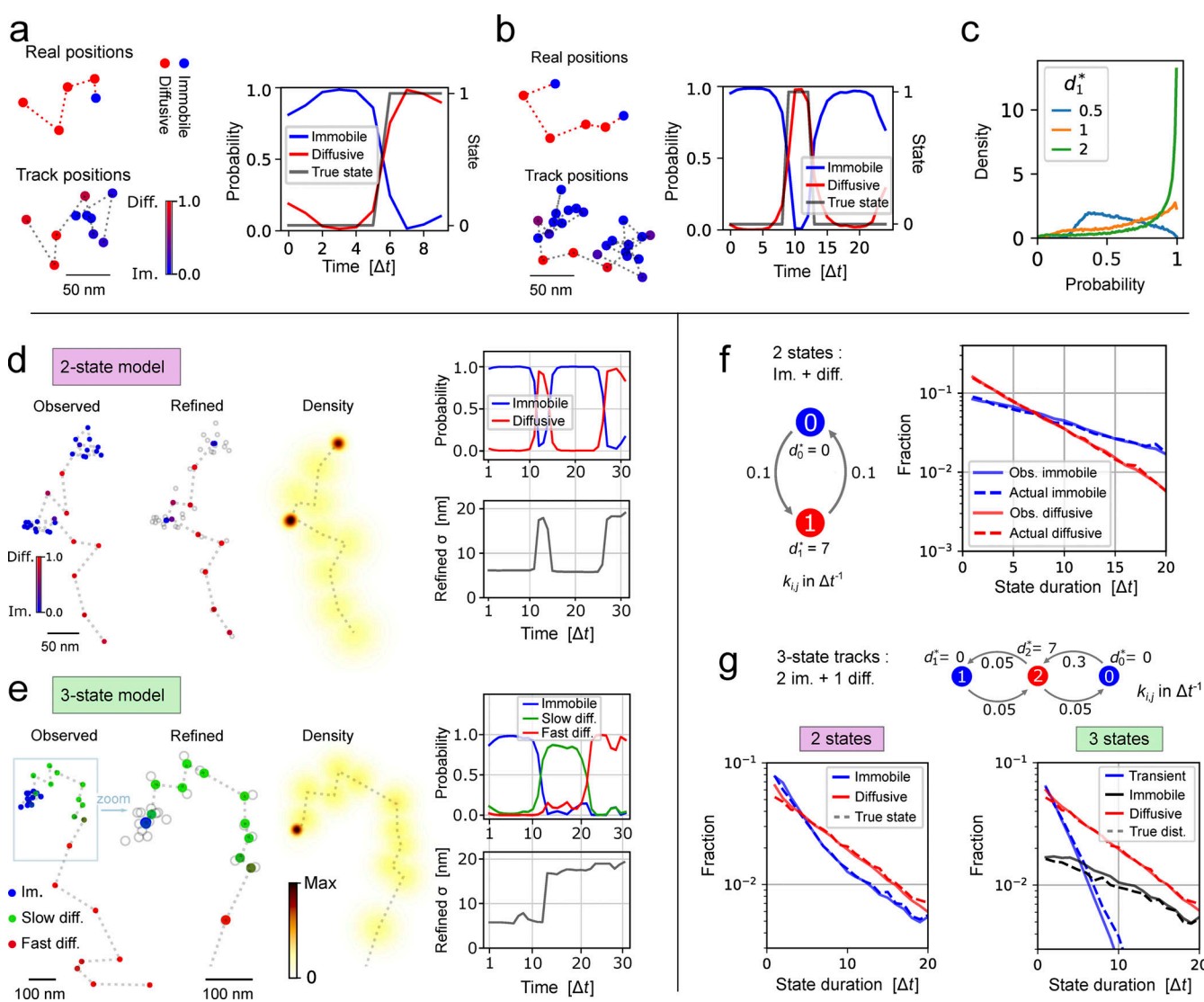

Figure 3. **ExTrack annotates and refines single-molecule positions and extracts state-duration distributions. (a and b)** Example tracks from simulations of immobile and diffusive ($d_1^* = 2$) molecules with symmetric transition rates of 0.1 $\Delta t^{-1}$ with short tracks of 10 positions (a) or long tracks of 25 positions (b). Top: real positions with states in colors; bottom: noisy positions with probabilities (color bar), right: state probabilities along time. **(c)** Distribution of the probability to be diffusive at the time points where the particles are actually diffusive. Similar simulations than in panel a for different $d^*$. **(d and e)** Position refinement module: Examples of two-state track (d) and three-state track (e). From left to right: Observed track and associated states probabilities; refined positions (colored) and observed positions in gray; probability density map of the consecutive positions. Top right: State probabilities as a function of time. Bottom right: SD of the probability density of refined positions. Simulation parameters: $d_0^* = 0$, $d_1^* = 5$, $k_{01} = k_{10} = 0.1 \Delta t^{-1}$ (d); $d_0^* = 0$, $d_1^* = 1.5$, $d_2^* = 5$, all $k = 0.05$ $\Delta t^{-1}$ (e). **(f)** Histogram module: State-duration histograms of tracks of at least 21 positions for the indicated two-state model. Dashed lines: Distributions from ground truth. **(g)** Same as f for three-state tracks with two immobile states. Left: ExTrack fit assuming a two-state model; Right: ExTrack fit assuming a three-state model.

reveals two sub-populations, even though ExTrack considers a two-state HMM model. Our approach thus indicates the presence of a third state, as confirmed by the exponential distributions of state durations after fitting the data to a three-state model (Fig. 3 g).

The histogram approach is also relevant when the transitions are non-Markovian, for example if transition rates are spatially dependent (Mahmutovic et al., 2012; Laurent et al., 2019) or if states have minimum durations. In summary, the histogram module can help identify hidden states or states with non-Markovian transitions and thus guide model choice.

## Application of ExTrack to experimental tracks of bacterial envelope proteins

To test our approach on experimental data, we used TIRF microscopy to track single GFP (monomeric super-folder-GFP) fusions to two bacterial membrane proteins in *Escherichia coli*, each involved in one of the two major pathways of cell-wall synthesis.

First, we studied the cell-wall-inserting penicillin-binding protein PBP1b (Video 1), which was previously described to reside in immobile or diffusive states (Cho et al., 2016; Vigouroux et al., 2020). However, transition rates and potentially hidden

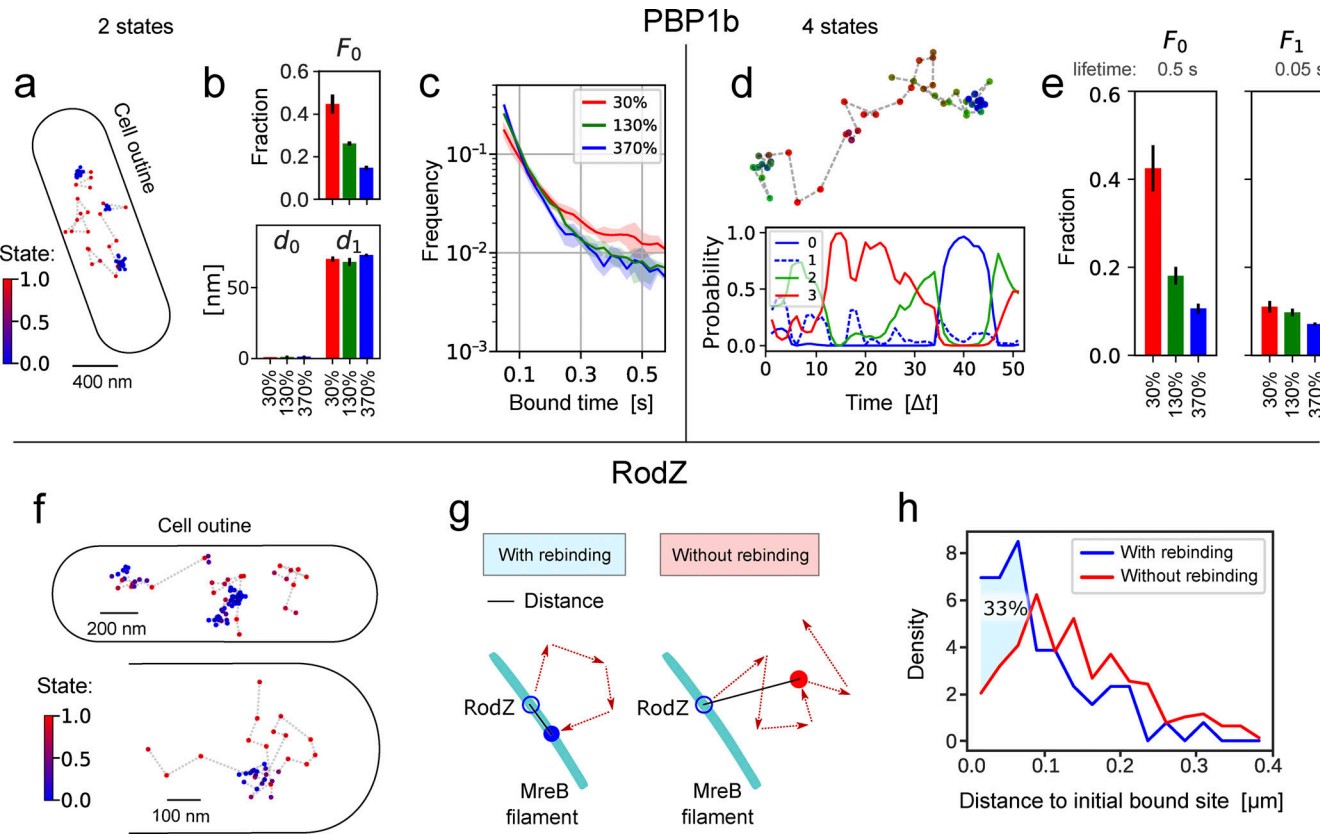

Figure 4.    **Characterizing PBP1b and RodZ motion. (a–e)** Analysis of GFP-PBP1b tracks (time step 25 ms) using ExTrack with two (a–c) or four states (d and e). **(a)** GFP-PBP1b track (130% expression level). Color bar: Probability of diffusion. **(b)** Diffusion lengths and fractions from two-state parameter fitting. (three replicates, each with >17,000 tracks of at least three positions, with average lifetimes from 6.3 to 7.5 positions). **(c)** State-duration histograms of PBP1b tracks of at least 21 positions (>600 tracks per replicate with average lifetimes from 28.5 to 32 positions), using global parameters from fitting a two-state model (a and Fig. S5). **(d)** Example PBP1b track (from 130% expression level) with associated state probabilities along time (first position on the left). **(e)** Fractions from four-state parameter fitting to the same datasets used in a. **(f–h)** Analysis of GFP-RodZ molecules (40 ms frame times). **(f)** RodZ tracks with overlapping binding sites. Color bar: Probability of diffusion. **(g)** Cartoon illustrating the rebinding of diffusive RodZ molecules to extended MreB-actin filaments in two dimensions. Blue solid line indicates distance between initial position and position after four diffusive steps for molecules rebinding (left) or continuing diffusion (right). **(h)** Histograms of distances between initial bound site and the site after four diffusive steps. Tracks rebind in closer vicinity to the initial binding site 33% more often than expected in case of random motion. Mann–Whitney U test: P value = 1.6e−6. See Materials and methods, Computational analysis... for details. Error bars and shaded regions: SDs between replicates.

states remain unknown. When assuming a two-state model, ExTrack indeed reveals an immobile and a diffusive fraction (Fig. 4 a), with the immobile fraction increasing with decreasing expression level (Fig. 4 b) as expected (Vigouroux et al., 2020). However, distributions of state durations obtained through the histogram module suggest the presence of at least two immobile populations with distinct unbinding rates (Fig. 4 c). Since applying ExTrack assuming a three-state model revealed one immobile and two diffusive states (rather than two immobile states, Fig. S5 a), we also applied ExTrack with a four-state model (Fig. 4, d and e; and Fig. S5 a). The four-state model confirmed two diffusive states and two immobile states: Among the immobile states we found a long-lived state (lifetime of around 0.5 s) that is highly dependent on expression level (Fig. 4 e), likely reflecting enzymatically active PBP1b, and a short-lived state with a lifetime of about 50 ms, likely reflecting non-specific associations with the cell wall. Thus, PBP1b displays rapid transitions between at least four different states.

Next, we investigated the motion of RodZ, a trans-membrane protein that physically links cytoplasmic MreB-actin filaments to a multi-enzyme complex that inserts new peptidoglycan while continuously moving around the cell circumference over minutes (Van Teeffelen et al., 2011; Morgenstein et al., 2015; Van Den Ent et al., 2010). Here, we studied the motion of GFP-RodZ on short time scales of seconds, where continuously moving complexes appear as immobile (Video 2). Assuming a two-state model, the fitting module reveals that 75% of RodZ molecules are immobile (Fig. S5 b based on rates), with a lifetime of about 0.7 s. This time scale is much shorter than the minute-long lifetime of the rod complex (Özbaykal et al., 2020; Cho et al., 2016; Van Teeffelen et al., 2011) demonstrating that a majority of immobile RodZ molecules is not stably associated. Instead, these molecules might transiently bind the MreB-actin cytoskeleton. Interestingly, RodZ molecules seem to often unbind and rebind in very close vicinity from the initial binding site (Fig. 4 f). Such behavior would be expected if RodZ could bind anywhere along extended MreB filaments, since filaments constrain

diffusion in two dimensions (Fig. 4 g). To test whether proximal rebinding occurs more often than randomly, we compared tracks that were initially bound, then diffusive for four steps, and then either rebound or remained diffusive (Fig. 4 g). Short distances were indeed over-represented among rebinding molecules compared to molecules that remained diffusive (Fig. 4 h). This behavior contrasts with PBP1b, which appears to bind to random sites (Fig. S5 e). The annotation module of ExTrack thus allows us to identify spatial patterns of molecule binding that can be responsible for non-Markovian binding (Mahmutovic et al., 2012).

## Discussion

In summary, ExTrack provides a suite of robust tools to characterize single-particle tracks, extracting global model parameters, state probabilities at every time point, refined positions, and histograms of state durations, even if tracks are noisy, transitions are rapid, and tracks deviate from idealized model assumptions.

In tracking experiments, a major challenge is to identify the relevant number of immobile and diffusive states. Multiple previous methods obtain this number automatically (Persson et al., 2013; Meent et al., 2013; Smith et al., 2019; Lindén and Elf, 2018; Karslake et al., 2021). However, at least some of these approaches have been reported to overestimate the number of states (Lindén and Elf, 2018; Vink et al., 2020a; Smith et al., 2019). In more recent approaches, a high number of states are fixed, followed by aggregation into one aggregated immobile state and one aggregated diffusive state, based on a user-defined diffusion-coefficient threshold (Metelev et al., 2022). Here, we propose an alternative and iterative approach to complex tracking data: Data are initially fit to a coarse-grained two- or three-state model that can subsequently be expanded depending on desired variables and fitting results. For example, if one is predominantly interested in the exchange between immobile and diffusive molecules but not in the presence of multiple diffusive states, we propose a coarse-grained two-state or an aggregated three-state model that reliably predicts immobile-diffusive transitions, even if diffusion coefficient is variable or if molecules transition between different diffusive states (Fig. 2 b and Fig. S3 b). At the same time, ExTrack can also distinguish multiple diffusive states explicitly (Fig. 1 e). Additionally, the distribution of immobile state durations can reveal the presence of multiple immobile fractions, which can then motivate the increase of the number of states.

The capacity of ExTrack to work with noisy single-molecule tracks is based on the explicit consideration of all sequences of states within a sliding window when computing the probability of every track, while states outside the sliding window are taken into account through averaging to limit computation time. In the future, this versatile principle can be extended to capture different and more complex dynamics, for example by considering persistent motion (Pierobon et al., 2009; Monnier et al., 2015), anomalous diffusion (Chen et al., 2021), and spatial maps of diffusion coefficients or states (El Beheiry et al., 2015).

## Materials and methods

In the following sections, we will describe the ExTrack method with its four different modules: the fitting module, the annotation module, the position-refinement module, and the histogram module. Subsequently, we will describe implementation and computational time, the generation of computationally simulated data sets, the interpretation of results by vbSPT, and the experimental methods.

### ExTrack fitting module
#### Introduction

ExTrack fits a multi-state diffusion model to noisy single-particle tracking data. We assume that tracks come about according to a continuous-time Markov model, where molecules transition randomly between $N$ diffusive states at rates $k_i$. As long as molecules reside in state $i$, they undergo Brownian diffusion with diffusion coefficient $D_i$. Additionally, observed positions $c_i$ are displaced from real positions $r_i$ according to a Gaussian distribution $f_\sigma(k_i - r_i)$, where the SD equals the localization error $\sigma$. Here and in the following, $f_x(y)$ generally denotes a Gaussian distribution of SD $x$. The $N$-state diffusion model is thus characterized by the parameters $\theta = (\sigma, D_i, F_i, k_{i,j})$ for all states $i, j \in (1, ..., N)$. Here, $F_i$ are the fractions of molecules residing in state $i$ at the first position of the track. Later, we will introduce additional parameters for additional spatial dimensions and for the treatment of non-constant track lengths.

Parameters are estimated based on an MLE approach, which, in turn, is based on accurately computing $f_C(C \mid \theta)$, the probability density of observing a track of positions $C = (c_1, c_2, ..., c_n)$. The likelihood of the parameters given the data ($\theta \mid$ all $C$) then equals the product of $f_C(C \mid \theta)$ for all tracks $C$. By maximizing this function, we can find $\theta^*$, the optimal estimator of the underlying parameters. Optimal parameters $\theta^*$ are found by MLE using the Powell method. ExTrack also allows to fix individual or multiple parameters. This generally speeds up the fitting process and reduces variations in the remaining parameters. In this realm, we also found that fixing the localization error to a slightly wrong value has little impact on the fitting of the other parameters as long as it does not deviate by more than about 20–30%. Here and in the following, we treat localization error as a model parameter, but the user can also provide spot-specific localization errors based on photon counts (Thompson et al., 2002).

In the following sections, we will first compute $f_C$. This calculation is presented in one spatial dimension (1D). However, the model is easily extendable to 2D or 3D due to the independence of the displacements and localization error in each axis, as we will see below.

#### Parameter fitting based on the probability distribution of observed positions

Tracks are generally described by their sequence of observed positions $C = (c_1, c_2, ..., c_n)$. Those positions come about based on the sequence of physical molecule positions $R = (r_1, r_2, ..., r_n)$, which, in turn, are the stochastic result of the sequence of diffusive states $B = (b_1, b_2, ..., b_n)$. For a given track $C$, the probability density function $f_c$ can be calculated from $f_{C,B,R}$, the joint

probability density function of the observed positions $C = (c_1, ..., c_n)$, the real (physical) molecule positions $R = (r_1, ..., r_n)$, and the time-dependent diffusion states $B = (b_1, ..., b_n)$, by integration over all the possible values of $R$ and by summation over all the possible values of $B$:

$$f_C(C|\theta) = \sum_B f_{C,B}(C, B \mid \theta) = \sum_B \int_R f_{C,B,R}(C, B, R \mid \theta) dR \quad (1)$$

where we defined the joint probability density function $f_{C,B}(C, B \mid \theta)$ of having $C$ and $B$ given $\theta$. The joint probability density $f_{C,B,R}$ can be decomposed into a product of three terms: the a priori probability of $B$, the probability density of the physical displacements $f_{R|B}$, and the probability density of the distances between real position and observed positions $f_{C|R}$, respectively:

$$f_{C,B,R}(C, B, R \mid \theta) = P(B \mid \theta) \, f_{R|B}(R \mid B, \theta) \, f_{C|R}(C \mid R, \theta). \quad (2)$$

Here, the a priori probability of the sequence of states $P(B \mid \theta)$, which we refer to as $\beta$ for brevity, results from the Markovian processes of transitioning between states (Cox and Miller, 2017). $\beta$ is obtained as

$$\beta = P(B \mid \theta) = F_{b_1} \prod_{i=1}^{n-1} p_{b_i, b_{i+1}}, \quad (3)$$

where $F_{b_1}$ indicates the fraction of molecules in state $b_1$ at time point 1, and where $p_{b_i, b_{i+1}}$ indicates the probability to transition from state $b_i$ at time point $i$ to $b_{i+1}$ at time point $i+1$. The transition probabilities can be computed from the continuous-time transition rates (see Approximating continuous transitions...). The initial fraction $F_{b_1}$ can either be an independent parameter or constrained by transition rates at steady state (for a two-state model, $F_0 = k_{10}/(k_{10} + k_{01})$). $f_{R|B}$ is the probability density function of real positions $R$ given the sequence of states $B$ and $\theta$. $f_{C|R}$ is the probability density function of the sequence of observed positions $C$ given the real positions $R$ and $\theta$.

$f_{C|R}$ can be expressed as a product of Gaussian distributions with SD equal to $\sigma$:

$$f_{C|R}(C \mid R, \theta) = \prod_{i=1}^{n} f_\sigma(c_i - r_i).$$

Next, we express $f_{R|B}(R \mid B, \theta)$ as a product of Gaussians:

$$f_{R|B}(R \mid B, \theta) = \frac{1}{l} \prod_{i=1}^{n-1} f_{\delta_i}(r_{i+1} - r_i). \quad (4)$$

Here, $l$ is the length of the space of real positions. Without any prior on $R$, we consider the limit $l \to \infty$. However, since $l$ only appears as a constant prefactor, we can ignore it in the calculation of the log likelihood. According to previous suggestions, the width of the distribution $\delta_i$ should equal the diffusion length corresponding to the current state $b_i$ (Das et al., 2009). However, this discretization of the continuous-time Markov process introduces a bias toward diffusive motion. This is easily illustrated in case of an immobile-diffusive model: There, a particle initially immobile starts moving before the first measured time point where the particle is observed to be diffusive. Similarly, it stops moving after the last time point, where it is observed to be diffusive. Then, a model assuming diffusion to only be dependent on the current hidden state will overestimate the time spent in the diffusive state by up to half a step in case of high

diffusion. Logically, this results in underestimating the binding rate (Fig. S2 c), immobile fraction, and diffusion length when transitions are frequent. To alleviate this issue, we assume transitions to occur at the middle of two time points. The SD of the probability density function $f_\delta$ then equally depends on states at each of the two subsequent time points, with

$$\delta_i = \sqrt{\left(d_{b_i}^2 + d_{b_{i+1}}^2\right)/2}.$$

This assumption effectively decreases the bias inherent to the discrete approximation of continuous tracks (Fig. S2 c). Later, we will also introduce sub-steps between time points that improve the approximation (see Approximating continuous transitions...).

Taking advantage of the expressions of $f_{C|R}$, and $f_{R|B}$, Eq. 2 becomes

$$f_{C,B,R}(C, B, R \mid \theta) = \frac{\beta}{l} \left[ \prod_{i=1}^{n-1} f_{\delta_i}(r_{i+1} - r_i) \, f_\sigma(r_i - c_i) \right] \times f_\sigma(r_n - c_n).$$

Inserting this expression into Eq. 1, we then integrate stepwise over all real positions $R = (r_1, ..., r_n)$. This allows us to use the recursion principle (Relich et al., 2016) to compute $f_{C,B}(C, B \mid \theta)$: The first step consists in integrating the two Gaussian distributions dependent on $r_1$ (displacement and localization error terms). This integration results in a Gaussian distribution $f_{s_1}(r_2 - c_1)$, of SD $s_1 = \sqrt{\sigma^2 + \delta_1^2}$ (constituting a convolution of two independent random variables with Gaussian distributions). For each of the next integrals over $r_i$, we integrate the product of three Gaussian distributions (for the random displacement $r_{i+1} - r_i$, the localization error $r_i - c_i$, and the previous term of the distribution $f_{s_{i-1}}$). The result of this integration can be expressed by a scalar $K_i$ times a Gaussian distribution $f_{s_i}$ according to

$$\int_{r_i \in R} f_{\delta_i}(r_{i+1} - r_i) f_\sigma(r_i - c_i) f_{s_{i-1}}(r_i - \mu_{i-1}) dr_i = K_i * f_{s_i}(r_{i+1} - \mu_i), \quad (5)$$

where $f_{s_i}$ is a Gaussian distribution of SD $s_i$ and mean 0. The SD $s_i$ and mean $\mu_i$ can be expressed depending on $s_{i-1}$ and $\mu_{i-1}$:

$$s_i = \sqrt{\frac{\delta_i^2 \sigma^2 + \delta_i^2 s_{i-1}^2 + \sigma^2 s_{i-1}^2}{\sigma^2 + s_{i-1}^2}},$$

$$\mu_i = \frac{\mu_{i-1} \sigma^2 + c_i s_{i-1}^2}{\sigma^2 + s_{i-1}^2},$$

$$K_i = \frac{\exp\left(-\frac{(c_i - \mu_{i-1})^2}{2(\sigma^2 + s_{i-1}^2)}\right)}{\sqrt{2\pi(\sigma^2 + s_{i-1}^2)}}. \quad (6)$$

The recursion process can then be summarized by the sequences $s_1 : s_{n-1}$ and $\mu_1 : \mu_{n-1}$ which depend on $C$, $B$, and $\theta$. At the last step (integration over $r_n$), we integrate the product of the two remaining Gaussian distributions: the previous term $f_{s_{n-1}}(r_n - \mu_{n-1})$ and the localization error term $f_\sigma(c_n - r_n)$ as described for the first step to compute the density function $f_{C,B}(C, B \mid \theta)$.

Finally, we compute the value of the probability density function of the observed track $f_C(C \mid \theta)$ as the sum of $f_{C,B}(C, B \mid \theta)$ over all possible $B$.

### Extension to 2D and 3D

Since diffusive motion is independent in each spatial dimension, the principle described above for one dimension can simply be extended to two or three dimensions by multiplication of independent distribution functions. For example, in 2D, the function $f_{C,B}(C, B \mid \theta)$ is simply replaced by the product $f_{C,B}(C_x, B \mid \theta)$ $f_{C,B}(C_y, B \mid \theta)$. In principle, each axis can have a different localization error and different diffusion lengths for each state. This is especially true for localization in the direction of the optical axis compared to the lateral axes. ExTrack, therefore, allows independent localization errors for each axis.

Alternatively, the user can also provide localization error for each peak, for example using the Cramer–Rao lower bound estimate (Ober et al., 2004; Lindén et al., 2017). Since these and other estimators might underestimate the true localization error, peak-wise localization estimates can also be implemented as scaling factors that are then assumed to be linearly related to the true localization error estimated by ExTrack.

### Using a window to reduce calculation time

This method has a number of operations which initially scales with $N^n$, where $N$ is the number of states and $n$ is the number of time points. This means the calculation time can become unrealistically long when analyzing long tracks. To alleviate this issue, we developed a window method to allow it to work with longer tracks in a reasonable time scaling with $nN^{m+1}$ ($m$ the window length of minimal value 1). For computational reasons, we advise to use a window length of 7 for two-state models, 5 for three-state models, and 3 or 4 for more states.

Here, we briefly motivate and describe the implementation of the window method: During the recurrence process described above, Eqs. 5 and 6, $f_s$ can be regarded as a density probability function of the position $r_{i+1}$ knowing the previous observed positions $c_1, \dots, c_i$ and states from positions 1 to $i + 1$. We realized that the current localization of a particle is very little affected by its state $m$ steps ago when $m \gg 1$. Thus, for a given track, two sequences of states varying only in their first state should give very similar $f_{s_i}$. The values $\mu_m$ and $s_m$ of these normal distributions should also be similar.

For example, if the track has been diffusive during at least one of the positions from steps $i$–$m$ to $i$, the current observed position $c_i$ is much more informative for the real position $r_{i+1}$ than the first observed position $c_1$. If the molecule has been immobile from position $i$–$m$ to $i$, all observed positions $c_{i-m}$ to $c_i$ are equally informative. However, even the past 5–7 positions are likely sufficient to predict the distribution $f_{s_i}$.

As we saw in Parameter fitting…, for a given sequence of states $B$, computing $f_{C,B}(C, B \mid \theta)$ is nothing but computing three sequences $s_1{:}s_n$, $\mu_1{:}\mu_n$, and $K_1{:}K_n$ until the last step where we simply have to compute $f_{s_n}(c_n)$.

In the recursion process, in case of a two-state model, we start by computing four values each for $s_2$, $\mu_2$, and $K_2$ that we will differentiate as $s_{2,(b_1,b_2)}$, $\mu_{2,(b_1,b_2)}$, and $K_{2,(b_1,b_2)}$, corresponding to the transition between the state $b_1$ at time point 1 and state $b_2$ at time point 2. For this recursion step, the four values of $s_2$, $\mu_2$, and $K_2$ arise from the following four combinations of states $(0, 0)$, $(0, 1)$, $(1, 0)$, $(1, 1)$. At $i = 3$, we get eight possible state combinations,

at $i = 4$ we get 16, etc. At step $m$, any sequence of states has a characteristic $\mu_{m,(b_1,b_2,\dots,b_{m+1})}$, $s_{m,(b_1,b_2,\dots,b_{m+1})}$, and $K_{m,(b_1,b_2,\dots,b_{m+1})}$. To limit the number of considered sequences to $2^m$, we can merge $\mu_{m,(0,b_2,\dots,b_{m+1})}$ and $\mu_{m,(1,b_2,\dots,b_{m+1})}$ to an average $\mu_{m,(*,b_2,\dots,b_{m+1})}$ (same for $s^2$ and $K$):

$$
\begin{aligned}
\mu_{m,(*,b_2,\dots,b_{m+1})} &= \alpha_{m,(0,b_2,\dots,b_{m+1})}\,\mu_{m,(0,b_2,\dots,b_{m+1})} \\
&\quad + \alpha_{m,(1,b_2,\dots,b_{m+1})}\,\mu_{m,(1,b_2,\dots,b_{m+1})}, \\
s^2_{m,(*,b_2,\dots,b_{m+1})} &= \alpha_{m,(0,b_2,\dots,b_{m+1})}\,s^2_{m,(0,b_2,\dots,b_{m+1})} \\
&\quad + \alpha_{m,(1,b_2,\dots,b_{m+1})}\,s^2_{m,(1,b_2,\dots,b_{m+1})}, \\
K_{m,(*,b_2,\dots,b_{m+1})} &= K_{m,(0,b_2,\dots,b_{m+1})} + K_{m,(1,b_2,\dots,b_{m+1})}.
\end{aligned}
$$

where $\alpha$ are the averaging weights according to the joint probability density of the observed position $c_1$ to $c_m$, and states $b_1$ to $b_{m+1}$ $f_{C,B}\big((c_1, c_2, \dots, c_m), (b_1, b_2, \dots, b_{m+1})|\theta\big)$. For brevity, we express this probability density as $P_{m,(b_1,b_2,\dots,b_{m+1})}$ in the following expression for $\alpha$:

$$
\begin{aligned}
\alpha_{m,(0,b_2,\dots,b_{m+1})} &= \frac{P_{m,(0,b_2,\dots,b_{m+1})}}{P_{m,(0,b_2,\dots,b_{m+1})} + P_{m,(1,b_2,\dots,b_{m+1})}}, \\
\alpha_{m,(1,b_2,\dots,b_{m+1})} &= \frac{P_{m,(1,b_2,\dots,b_{m+1})}}{P_{m,(0,b_2,\dots,b_{m+1})} + P_{m,(1,b_2,\dots,b_{m+1})}}.
\end{aligned}
$$

In this way, two sequences of states are merged (for example, the sequences starting with $[0,0,0,1,1,1]$ and $[1,0,0,1,1,1]$). We thus reduce the number of $\mu_m$, $s^2_m$, and $K_m$ from $2^{m+1}$ to $2^m$. By recursion of this principle over all steps from $m$ to $n-1$ we limit the computation time to $2^{m+1}$ for a two-state model, or, more generally, to $N^{m+1}$ for a $N$-state model. In the following section, we will introduce sub-steps between the discrete observation time points. Our approach is easily generalized to sub-steps by considering state vectors. Applying our approach with a window length $m = 5$–7, we observed similar functional dependencies of the likelihood on the model parameters $\theta$, allowing us to drastically speed up our method without losing accuracy.

### Approximating continuous transitions with a discrete model of one or multiple sub-steps per time frame

ExTrack fits data of a continuous-time process to a discrete-time Markov model. Without the introduction of sub-steps, ExTrack assumes that transitions can only happen once per time step. It then estimates transition probabilities per time step, which must be translated into transition rates $k_i$, that describe the continuous-time Markov model. For continuous-time Markov processes, transition probabilities can be converted into rates according to a simple relationship $P = e^{G\Delta t}$, where $P$ is the transition probability matrix, which contains the transition probabilities $p_{i,j}$ from state $i$ to $j$, and where $G$ is the generator matrix with elements $G_{i,j} = k_{i,j}$ for $i \neq j$ and $G_{i,i} = -\sum_{i \neq j} k_{i,j}$ (Pishro-Nik, 2016). Here, the transition probabilities $p_{i,j}$ allow the molecule to transition from state $i$ to $j$ through any number of intermediate states.

However, the implementation of this relation into ExTrack leads to a systematic overestimation of transition rates. The reason for this overestimation is found in our approximate representation of the distribution of physical displacements between time points (Eq. 4), which is based on the false

assumption that transitions can only occur at the middle of steps, contrary to the continuous-time nature of the underlying physical process. We found that this error could be compensated for by using a slightly different approximation for state transitions $p_{i,j} = 1 - \exp(-k_{i,j} \Delta t)$. In the limit of small $k_i$, $\Delta t$, this approximation asymptotically equals the exact expression $P = e^{G\Delta t}$, which also asymptotically equals $p_{i,j} = k_{i,j} \Delta t$. We found that ExTrack using the approximate relationship performs better in the case of two-state and three-state models for a large range of transition rates. However, ExTrack (Python version) also allows using the generator-matrix-based relationship if the user desires.

When transition rates are high (when $k \Delta t > 0.4$ for the two-state model), our method allows to sub-divide time steps into a number of $u$ sub-steps (where $u = 2$ corresponds to dividing each step into two). This allows ExTrack to account for multiple transitions and transition times that are different from the midpoints of time steps $\Delta t$. To take states at sub-steps into account, we introduce a new state vector $B = (b_{1,1}, b_{1,2}, ... b_{1,u} ... , b_{n-1,u}, b_{n,1})$ and new physical positions, that require integration according to Eq. 1. This integration is straightforward: The probability $\beta$ (Eq. 2) is simply replaced by the product of all state transitions between subsequent sub-positions. The probability distribution of real positions (Eq. 4) is replaced by the corresponding distribution of sub-positions. Since the physical displacement during $\Delta t$ is the sum of Gaussian random variables (the sub-displacements), the functional form of Eq. 4 (a product of $n-1$ Gaussians) can be maintained while replacing the SDs $\delta_i = \sqrt{d_i^2 + d_{i+1}^2}$ by $\delta_i = \sqrt{\sum_{k=1}^{u} \delta_{i,k}^2 / u}$, where $\delta_{i,k}$ are the corresponding diffusion lengths for the sub-steps.

The use of a window length $m$ will allow the user to do accurate computations for $m$ sub-steps. Thus, at a given $m$, the number of observed positions $c$ considered within the window equals floor($m,u$). To consider the same number of observed positions per window, one thus needs to increase the window length. A trade-off between number of states, window length, and number of sub-steps has to be found (see Implementation and computation time).

### Extension of ExTrack to consider a finite field of view

Tracks can terminate due to different reasons: photobleaching, diffusive molecules leaving the field of view, or molecules transiently not being detected. The process of leaving the field of view requires diffusive motion. Observation of long-lived molecules within a finite field of view can thus show a bias toward non-moving or slowly moving molecules. An extension of ExTrack can take this bias into account by explicitly modeling the probability of track termination. We consider two contributions to track termination: first, a constant termination probability $p_K$, which is independently of the motion state. This probability summarizes photobleaching and the probability to not detect a molecule, for example because of low signal-to-noise ratio; second, a probability of leaving the field of view (or observation volume) $p_L$ that depends on the diffusion length and the dimensions of the field of view. In the case of a cytoplasmic particle tracked through epi-fluorescence or confocal microscopy, the monitored length is the depth of field (or focal depth). In case

of a membrane protein moving around a cylindrical cell imaged in TIRF microscopy, the monitored length is a fraction of the cell diameter (Fig. 2, c and d).

In principle, $p_L$ can be calculated depending on the position of the molecule with respect to the boundaries of the field of view. However, we decided to implement an approximate form of $p_L(\delta_i)$ that does not require this information and instead considers the position of the observed molecule as random inside the field of view. Within this approximation, the probability of leaving the field of view is given by:

$$p_L(\delta_i) = 1 - \int\limits_{x \in [0,l]} F\left(\frac{l-x}{\delta_i}\right) - F\left(\frac{-x}{\delta_i}\right) dx,$$

where $F(x)$ is the cumulative density function of the standard normal law.

We thus modify $f_{C,B,R}(C,B,R \mid \theta)$ in Eq. 2 by multiplication of the left-hand terms with the probability of observing a track of $n$ positions, which is given by:

$$\left(1 - p_L(\delta_i)\right)^{n-1}\left(1 - p_K\right)^{n-1}\left[p_K + (1 - p_K)p_L(\delta_n)\right].$$

### Annotation module

The annotation module allows to compute the probabilities to be in any state at any time point of all tracks. According to conditional probabilities and results from ExTrack fitting module, we can compute the probability of the sequences of states given the parameters $\theta$ for each track $C$:

$$P(B \mid C, \theta) = \frac{f_{C,B}(C, B \mid \theta)}{f_C(C \mid \theta)}.$$

For a given track $C$, at each time point $i$, the probability of the current state $b_i$ to be in state $s \in \{0, 1\}$ can then be computed by summing over all $B$ with $b_i = s$:

$$P(b_i = s \mid C, \theta) = \sum_B P(b_i = s \mid B, \theta)\frac{f_{C,B}(C, B \mid \theta)}{f_C(C \mid \theta)}.$$

The annotation module can also take advantage of the window approximation described in Using a window... to reduce computation time and make the computation tractable in case of long tracks. Since the annotation module does not require parameter fitting and thus many iterations, the window length can be chosen larger than for the fitting module. A large window length is also more important for precise state prediction than for accurate global parameter fitting.

### Position-refinement module

ExTrack can improve the estimation of molecule positions based on a track, in particular if molecules move slowly. Positions can be estimated by computing the probability density function of each real positions $r_i$ (Lindén et al., 2017). To do so, we compute $f_C(C \mid \theta)$ without integrating at position $r_i$. This results in a probability density function $f(r_i \mid C, \theta)$ (Fig. 3, d and e), which is a sum of Gaussian functions for each sequence of states. While this probability density can be obtained explicitly, it is much faster to obtain the expected value and SD of the density function. Those values are computed by averaging the parameters of the

Gaussian distributions associated with each sequence of states weighted by their respective probability. Like for the fitting method, the window method is applied (see Using a window...).

### Histogram module

Computing state-duration histograms allows to assess non-Markovian transition behaviors or to reveal multiple hidden states with different transition rates. For a given state, the resulting rate is then the sum of track-termination rate (bleaching, track termination due to low signal-to-noise ratio, leaving the focal plan) and the transition rates to other states. If the track-termination rate is low, the histogram allows to identify one or multiple transition rates (see, for example, Fig. 3, f and g). Picking only long tracks can help removing the contribution from bleaching.

ExTrack estimates the histograms $h_s$ for each state $s$:

$$h_s(i) = \sum_C \sum_B g(i, s \mid B) \frac{f_{C,B}(C, B \mid \theta)}{f_C(C \mid \theta)},$$

with $g(i, s \mid B)$ the number of sequences of $i$ consecutive time points of state $s$ in the sequence of states $B$. As long tracks have to be assessed and all states of $B$ kept in memory, the window method cannot be applied. We thus only keep the most likely $B$ (1,000 in Fig. 3, f and g).

### Implementation and computation time

ExTrack is available as a Python (Python 3) package (Simon, 2022). A version with the core functionality is also available as a TrackMate module (Ershov et al., 2022) on Fiji. The TrackMate implementation can fit data to a two-state model and annotate states according to the results from the fit or manually chosen parameters. It then allows interactive visualization of tracks colored with state probabilities for each displacement. We allow parallelization with GPU (cupy library; Python version only) or multiple CPUs (both Python and TrackMate versions).

As mentioned in Approximating continuous transitions..., a trade-off between number of states $N$, window length $m$, and the number of sub-steps $u$ has to be found for reasonable computation time. When running the ExTrack fitting module on a computer with Intel Core i7-9700 processor (10,000 tracks of 10 positions) for 200 iterations using a two-state model, a window length of $m = 2$, and no sub-step ($u = 1$) the analysis can be as fast as 20 s. For the dependency of computation time on numbers of states, sub-steps, and window length, see Fig. S1 c.

To save computation time, we recommend to initially run ExTrack with low values of $u$ and $m$ and then to increase $u$ if model predictions suggest high transition rates or $m$ for low predicted diffusion lengths. Specifically, we suggest making the following adjustments: If localization error is negligible, for instance if there is no immobile state and all $d_i > 2\sigma$, window length $m$ can be set to its minimal value of 1. Similarly, $m = 1$ should perform well if there is one immobile state and all diffusive states have large $d_i > 5$. In such cases, multiple sub-steps can be used at little computation cost. More generally, if predicted transition rates are larger than $0.4 \, \Delta t^{-1}$ but localization error is not negligible, we suggest increasing $u$ to 2 for most accurate estimates (Fig. S1 b vs. Fig. 1 f). In the hardest cases of small $d^* \lesssim 2$ and high transition rates ($\geq 0.4$), we recommend using $u = 2$ and $m \geq 8$.

### Computational simulation of tracks

To test the predictive power of the different methods, we conducted overdamped Brownian Dynamics simulations of tracks in two or three spatial dimensions with molecules transitioning randomly between the different states at discrete time points. To mimic a continuous-time Markov model for state transitions, we used a small time step $\tau = \Delta t/50 \ll 1/k_{i,j}$, where $k_{i,j}$ are the transition rates. Brownian Dynamics simulations were carried out by randomly drawing physical displacements in each spatial dimension from Gaussian distributions of SD $\sqrt{2D\tau}$, where $D$ is the diffusion coefficient corresponding to the diffusive state. An additional Gaussian distributed noise of SD $\sigma$ was added to simulate localization uncertainty.

Except if specified otherwise, we simulated 10,000 tracks of 10 time points with localization error $\sigma = 0.02 \, \mu m$, $\Delta t = 0.06$ s, $d_0 = 0 \, \mu m$, $d_1 = 0.1 \, \mu m$, bound fraction $F_0 = 0.5$, transition fractions per steps $k_{01} = k_{10} = 0.1 \, \Delta t^{-1}$, with infinite field of view and perfectly stroboscopic tracks. We also assumed that molecules reached steady state, that is, $F_0 \cdot k_{01} = (1 - F_0) \cdot k_{10}$.

To test the robustness of ExTrack and the other methods to more complex behaviors, we also simulated tracks with variations of localization error or diffusion coefficients, a finite field of view or physical confinement as follows (see Fig. 2 for illustrations):

Track-to-track variations of localization error (or diffusion coefficients) were simulated with localization error $\sigma$ (or diffusion coefficient $D$) following $\chi^2$ distributions of given coefficients of variation and mean $0.02 \, \mu m$ (or $0.25 \, \mu m^2 \, s^{-1}$ for $D$), $\Delta t = 0.02$ s. Models with multiple diffusion states were simulated as continuous-time transitions with model parameters detailed in Table S2.

To simulate a finite field of view in two dimensions, we simulated tracks in a box that is infinite in one spatial dimension ($y$) and finite in the other dimension ($x$) with size $3l$, where $l$ is the size of the field of view. All tracks or part of tracks that fall into the field of view are considered for further analysis. A single particle can thus result in several tracks if leaving the field of view and coming back. A finite field of view in three spatial dimensions was simulated analogously: The simulation box is infinite in $x$- and $y$-directions, while the box has periodic boundary conditions in the $z$-direction.

To simulate physical confinement, we considered tracks to move within a square area of indicated side length, using reflecting boundary conditions.

We also simulated tracks with one bound state and several intractable diffusive states to determine whether ExTrack could accurately estimate global binding and unbind with a coarse-grained three-state model. In Fig. S3, we show the results obtained from simulations of tracks with one bound state and five diffusive states of $d^* = [2,3,4,5,6]$. transition rates (in $\Delta t^{-1}$): $k_{01} = 0.03$, $k_{02} = 0.02$, $k_{03} = 0.06$, $k_{04} = 0.04$, $k_{05} = 0.05$, $k_{10} = 0.04$, $k_{12} = 0.05$, $k_{13} = 0.1$, $k_{14} = 0.07$, $k_{15} = 0.04$, $k_{20} = 0.19$, $k_{21} = 0.01$, $k_{23} = 0.04$, $k_{24} = 0.06$, $k_{25} = 0.1$, $k_{30} = 0.02$, $k_{31} = 0.03$, $k_{32} = 0.05$, $k_{34} = 0.05$, $k_{35} = 0.1$, $k_{40} = 0.1$, $k_{41} = 0.04$, $k_{42} = 0.04$,

$k_{43}$ = 0.01, $k_{45}$ = 0.05, $k_{50}$ = 0.15, $k_{51}$ = 0.05, $k_{52}$ = 0.01, $k_{53}$ = 0.02, $k_{54}$ = 0.05. The 10,000 tracks of 10 time points were then fitted to a three-state model, window length = 6.

## Comparison to vbSPT

To compare our results with vbSPT, we fixed the number of states to two so both algorithms performed exactly the same task. vbSPT does not consider localization error but a metric that we will call $u$. In case of pure diffusion, $u = D \cdot \Delta t$ but in case of immobile particle with localization error $u = \sigma^2/2$ in principle. We can thus infer $\sigma$ and $D$ according to $\sigma = \sqrt{2 \cdot u_0}$ and $D = (u_1 - u_0)/\Delta$.

## Computational analysis of molecule rebinding

To assess the propensity of RodZ molecules to rebind in close vicinity of their initial binding site, we first annotated tracks using parameters obtained from the ExTrack fitting module (three biological replicates). We considered the 16 first time points of tracks of at least 16 time points (pooling the tracks of the three replicates). Among tracks labeled as initially immobile for at least three time points ($p_{immobile}$ > 0.5) then diffusive for four time points (with at least three time points of probability $p_{diffusive}$ > 0.7), we grouped tracks into two sub-groups, the ones rebinding right after and the ones which continue to diffuse for at least one more time point. The histograms represent the distributions of distances between initially bound position and the position at the fourth time point after unbinding. If molecules were to rebind at random locations, the distributions of the distribution of distances for rebinding particles should be the as for particles, which continue to diffuse. Neither PBP1b (Fig. S5 e) nor tracks obtained from immobile fluorescent beads (of similar signal-to-noise ratio) showed any significant rebinding, which excludes wrong conclusions on RodZ data due to mis-annotations.

## Cell cultures

We used the IPTG-inducible GFP-RodZ strain FB60(iFB273; Δ*rodZ*, Plac:: *gfp-rodZ*) by Bendezú et al. (2009) and the GFP-PBP1b-containing strain AV51 (*msfgfp-mrcB*, Δ*mrcA*; Vigouroux et al., 2020). Cells were grown overnight at 37°C (shaking) in LB medium and then washed and diluted at least 1:1,000 in M63 minimal medium (Miller, 1,972) supplemented with 0.1% casamino acids, thiamine (5 × 10⁻⁵%), glucose (0.2%), and MgSO4 (1 mM) and grown for 6 h to early exponential phase (maximum OD600 of 0.1) at 30°C (shaking). Cells were then spread on an agar pad made from the same M63 media as described above. RodZ production was induced with 100 μM IPTG. In the strain AV51, CRISPR repression of msfGFP-PBP1b is induced with 100 ng/ml of anhydro-tetracycline (Acros Organics). When necessary, strains were supplemented with kanamycine (50 μg/ml) or carbenicillin (100 μg/ml) during overnight cultures. Biological replicates result from independent cultures starting from separate colonies.

## Single-particle tracking of msfGFP-PBP1b and GFP-RodZ proteins

Cells were all positioned in the same focal plan in between an agar pad (1%) and a coverslip to be imaged in TIRF microscopy. Coverslips were cleaned by 60 min sonication in saturated KOH

solution followed by two washing steps (15 min sonication in milli-Q water). Single-particle tracking of GFP-PBP1b was performed with a custom-designed fluorescence microscope based on an ASI Rapid Automated Modular Microscope System, equipped with a 100× TIRF objective (Apo TIRF, 100×, NA 1.49, Nikon), Coherent Sapphire 488–200 laser, and a dichroic beamsplitter (Di03-R488/561-t3-25 × 36, Semrock). Excitation was controlled with an acousto-optic tunable filter (AA Optoelectronics) through an Arduino (15 ms light exposure per frame). Images were acquired using an Andor iXon Ultra EMCCD camera with an effective pixel size of 130 nm. Image acquisition was supervised with MicroManager.

Tracks were built from movies using TrackMate with LoG peak detection (estimated blob diameter of 0.5 μm, quality threshold of 15, with subpixel localization). Then, peaks were linked into tracks using the "Simple LAP Tracker" (max distances of 0.4 μm, and "gap" of 2).

Data analysis with ExTrack was restricted to tracks with at least three positions. For the longest tracks, only the first 50 positions were analyzed.

## Online supplemental material

Fig. S1 shows ExTrack parameter fitting. Fig. S2 shows error on two-state model parameters for different methods. Fig. S3 shows robustness of ExTrack to biases due to wrong model assumptions. Fig. S4 shows capacity of the annotation module. Fig. S5 shows complementary results for GFP-PBP1b and GFP-RodZ tracks. Video 1 shows msfGFP-PBP1b single-particle tracking. Video 2 shows GFP-RodZ single-particle tracking. Table S1 shows using ExTrack to fit parameters of a three-state model to different simulations of three-state data with qualitatively different types of transitions. Table S2 shows two-state and three-state fits of tracks from simulated particles either in immobile state (state 0) or in one of five diffusive states (states 1–5).

## Data availability

Single-particle tracks, ExTrack results, and selected movies of the experiments corresponding to Fig. 4 are available on the Zenondo repository (https://doi.org/10.5281/zenodo.7548793).

## Acknowledgments

We thank the former lab members Andrey Aristov for setting up the TIRF microscope, Antoine Vigouroux for the GFP-PBP1b strains, and Gizem Özbaykal for her guidance for experiments. We also thank Felipe Bendezú (Umeå University, Sweden) and Piet De Boer (Case Western Reserve University, Cleveland, OH, USA) for the GFP-RodZ fusion.

This work was supported by the European Research Council under the European Union's Horizon 2020 research and innovation program (Grant Agreement No. 679980) to S. van Teeffelen, the French Government's Investissement d'Avenir program Laboratoire d'Excellence "Integrative Biology of Emerging Infectious Diseases" (ANR-10-LABX-62-IBEID, S. van Teeffelen), France BioImaging (ANR-10-INBS-04, J.-Y. Tinevez), the Mairie de Paris "Emergence(s)" program to S. van Teeffelen, a Natural Sciences and Engineering Research Council of Canada

Discovery Grant to S. van Teeffelen, a Fonds de Recherche du Québec - Santé Salary Fellowship to S. van Teeffelen, as well as support from the Volkswagen Foundation to S. van Teeffelen. Open Access funding provided by Université de Montréal.

Author contributions: F. Simon: Conceptualization, Methodology, Software, Validation, Formal analysis, Investigation, Writing. He conceived and developed ExTrack, carried out all experiments, conducted the analysis, and wrote the paper. J.-Y. Tinevez: Software. He ported ExTrack to the Fiji plugin Track-Mate. S. van Teeffelen: Conceptualization, Methodology, Writing, Supervision, Project administration, Funding acquisition. All authors reviewed the final manuscript.

Disclosures: The authors declare no competing interests exist.

Submitted: 14 August 2022

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

# Supplemental material

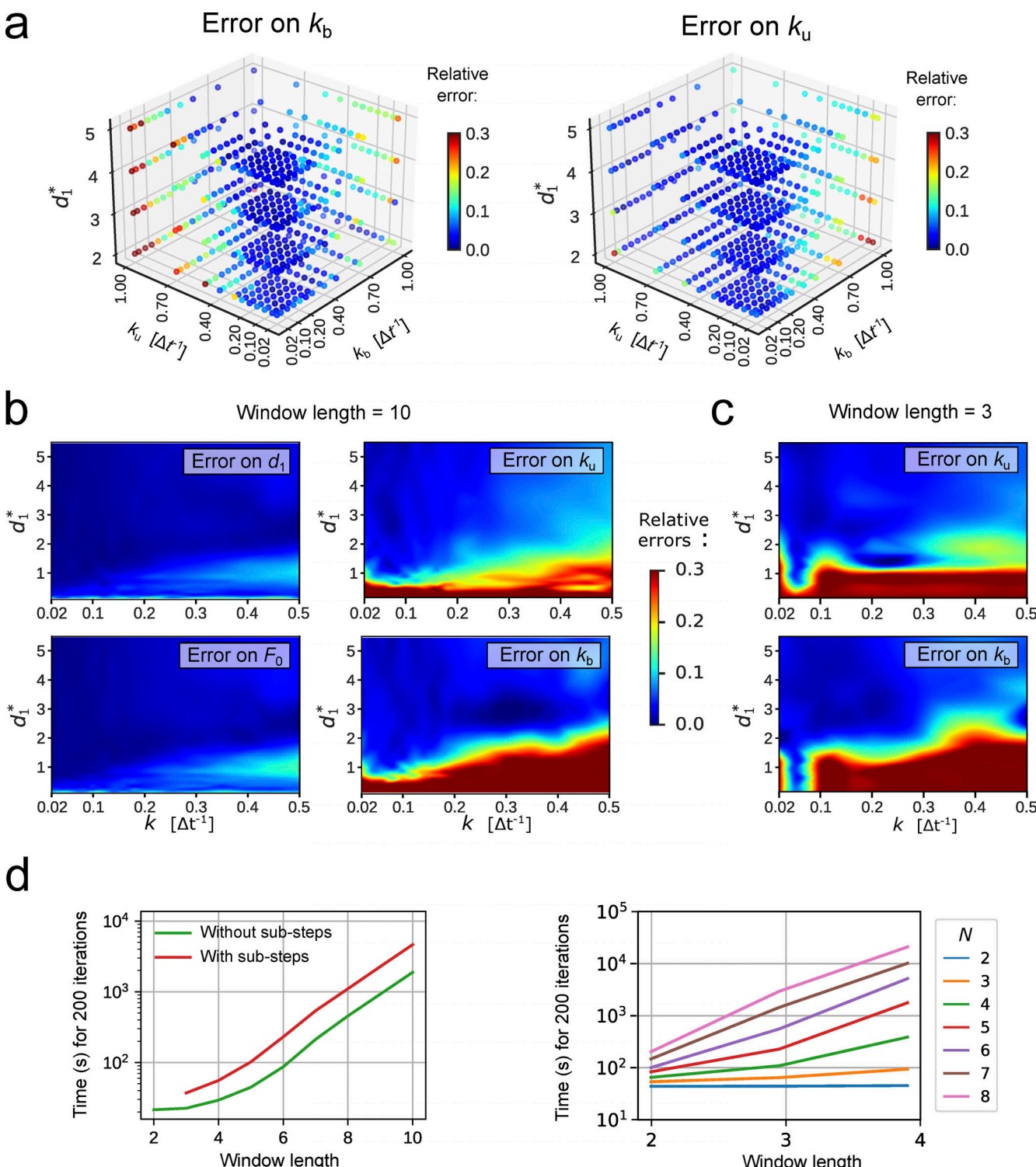

**Figure S1.  ExTrack parameter fitting. (a)** 3D maps of the mean error on extracted parameters from simulations similar to those in Fig. 1, of two-state models with one immobile and one diffusive state as a function of diffusion length $d^*$, unbinding rate $k_u$, and binding rate $k_b$. Errors are obtained from five replicates. Errors are indicated as absolute or relative errors, as indicated. ExTrack settings: no sub-steps, window length = 7. **(b and c)** Heat maps of mean relative error on extracted parameters from the same simulations as in Fig. 1, but inferred with no sub-steps. ExTrack settings: no sub-steps, window length = 10 (b) or 3 (c). **(d)** Computation time of ExTrack fitting module applied to 1,000 tracks of 100 positions each with 200 iterations (typical number of iterations needed for the fit of a two-state model) depending on the window length $m$ (without sub-steps or with two sub-steps; left) or depending on the number of states $N$ (without sub-steps; right), using seven cores of a Intel Core i7-9700 processor. For details see Materials and methods, Implementation and computation time.

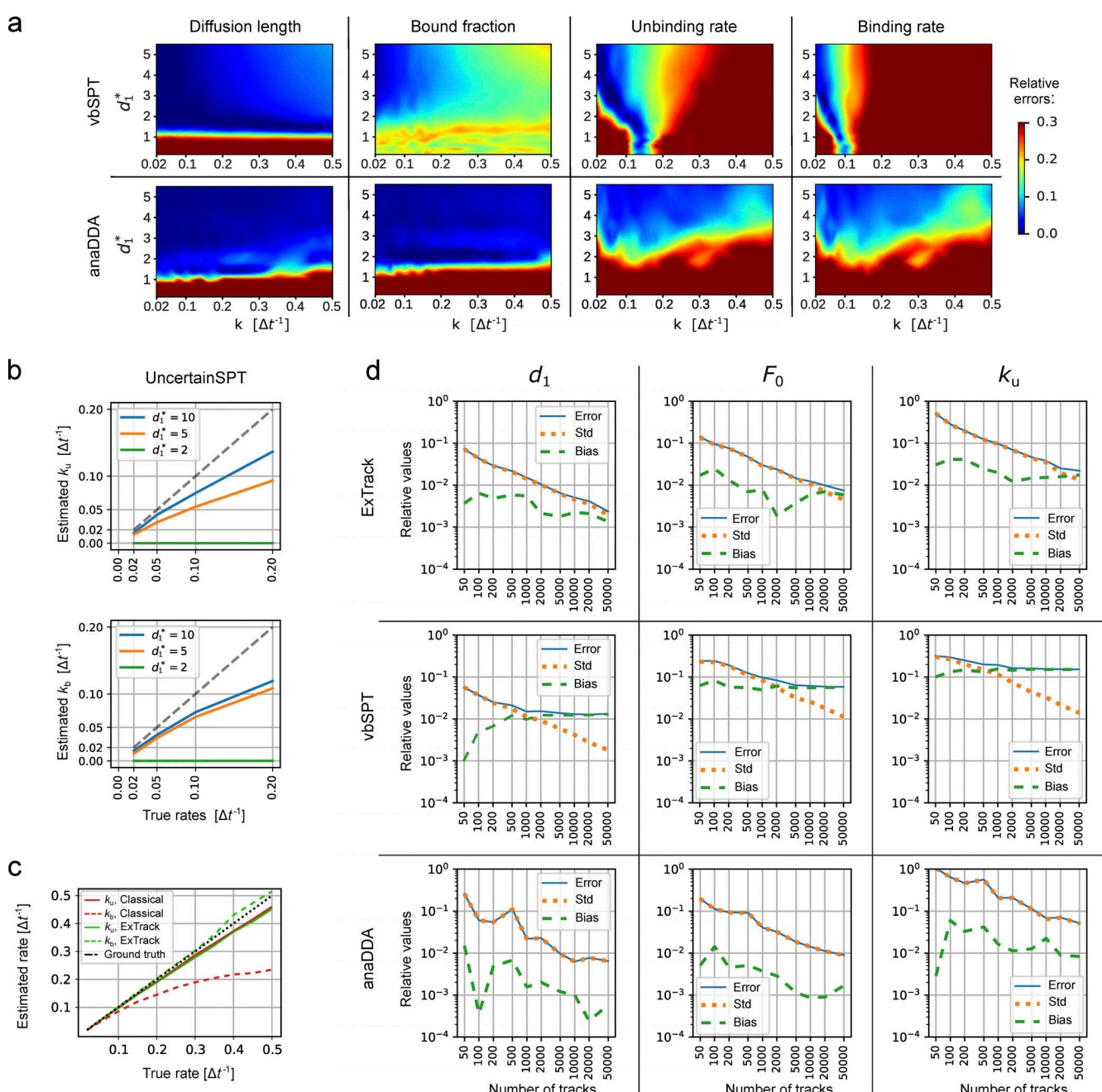

Figure S2. **Error on two-state model parameters for different methods. (a)** Heat maps of mean relative error on extracted parameters from the same two-state simulations as in Fig. 1 for vbSPT and anaDDA. In the case of anaDDA, 20 replicates were used to assess its heat maps instead of 10 due to its higher variability. **(b and c)** Plots of estimated transition rates as a function of the actual rates for a subset of the simulations in a. **(b)** Results from UncertainSPT for different diffusion lengths. Gray curve: Ideal model. **(c)** Results from a modified version of ExTrack with a time-discretization approach (labeled Classical), which assumes transitions to occur at time points of molecule observations, and with our approach (labeled ExTrack), which assumes transitions to occur at the middle of each time step (see Materials and methods, Parameter fitting...). Tracks simulated with $d_0^* = 0$, $d_1^* = 5$. **(d)** Error, SD, and bias of parameters predicted by ExTrack, vbSPT, and anaDDA depending on the number of tracks in case of two-state tracks (five positions per track, $k_u = k_b = 0.1\,\Delta t$, $d_0^* = 0$, and $d_1^* = 5$). The error (root mean square error) can be decomposed into bias (absolute value of the difference between the average estimate from all replicates and true parameter) and SD (std) of the estimated parameters. Error = $\sqrt{\text{bias}^2 + \text{std}^2}$. Obtained from 100 replicates. Here, all estimated values are relative to their true value. ExTrack settings: number of sub-steps = 2, window length = 10.

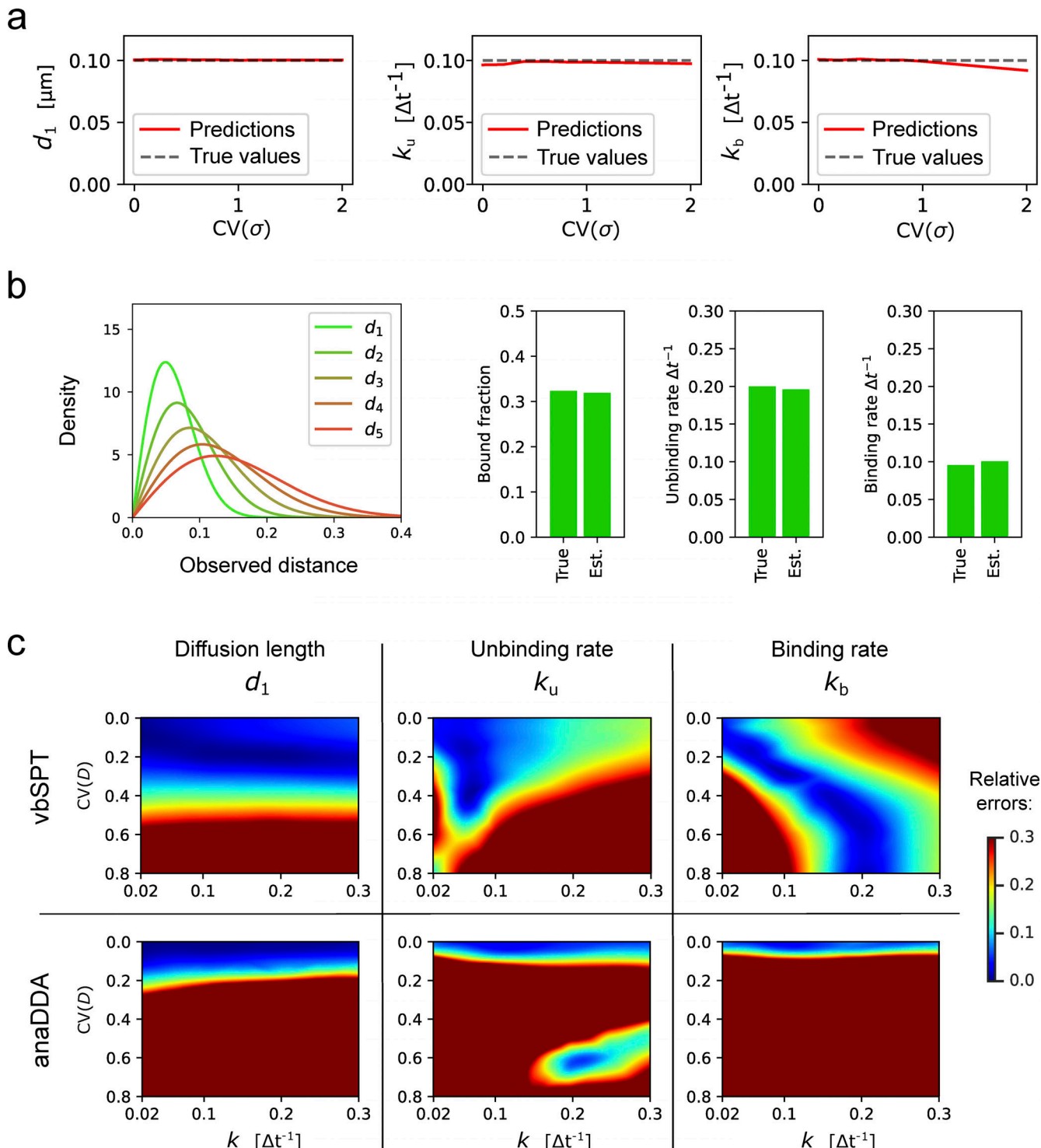

Figure S3. **Robustness of ExTrack to biases due to wrong model assumptions. (a)** Predictions of $d_1$, $k_u$, and $k_b$ in case of two-state parameter fits to two-state simulations with one immobile state and one diffusive state. Position at each time point show variable localization errors $\sigma$. Peak-wise localization errors were specified to the model. $\sigma$ followed a chi-square distributions re-scaled so the mean localization error equals 0.02 μm (for sample distributions, see inset of Fig. 2 b). Simulations with $d_1$ = 0.1 μm and $k = k_u = k_b = 0.1\Delta t^{-1}$. 10 replicates per condition. ExTrack settings: window length = 7, no sub-steps. **(b)** We considered tracks from simulated particles with one immobile state ($d_0^* = 0$) and five diffusive states with similar diffusion lengths of values 0.04, 0.06, 0.08, 0.1, and 0.12 μm (corresponding to $d^*$ from 2 to 6), transition rates were set to randomly picked values (See Materials and methods, Computation simulation of tracks for more details on the transition rates values). This model results in indistinguishable diffusive tracks. Left: Distribution of displacements (for each dimension) of the five diffusive states. Right: Bar plots of true and estimated parameters obtained from fitting to a three-state model followed by aggregation of the diffusive states and computation of the resulting parameters. Here, the fractions are the global fractions computed from rates. See Table S2 for other results. **(c)** Heatmaps of relative errors on $d_1$, $k_u$, and $k_b$ with variable diffusion coefficient following the same protocol than in Fig. 2 b for vbSPT and anaDDA.

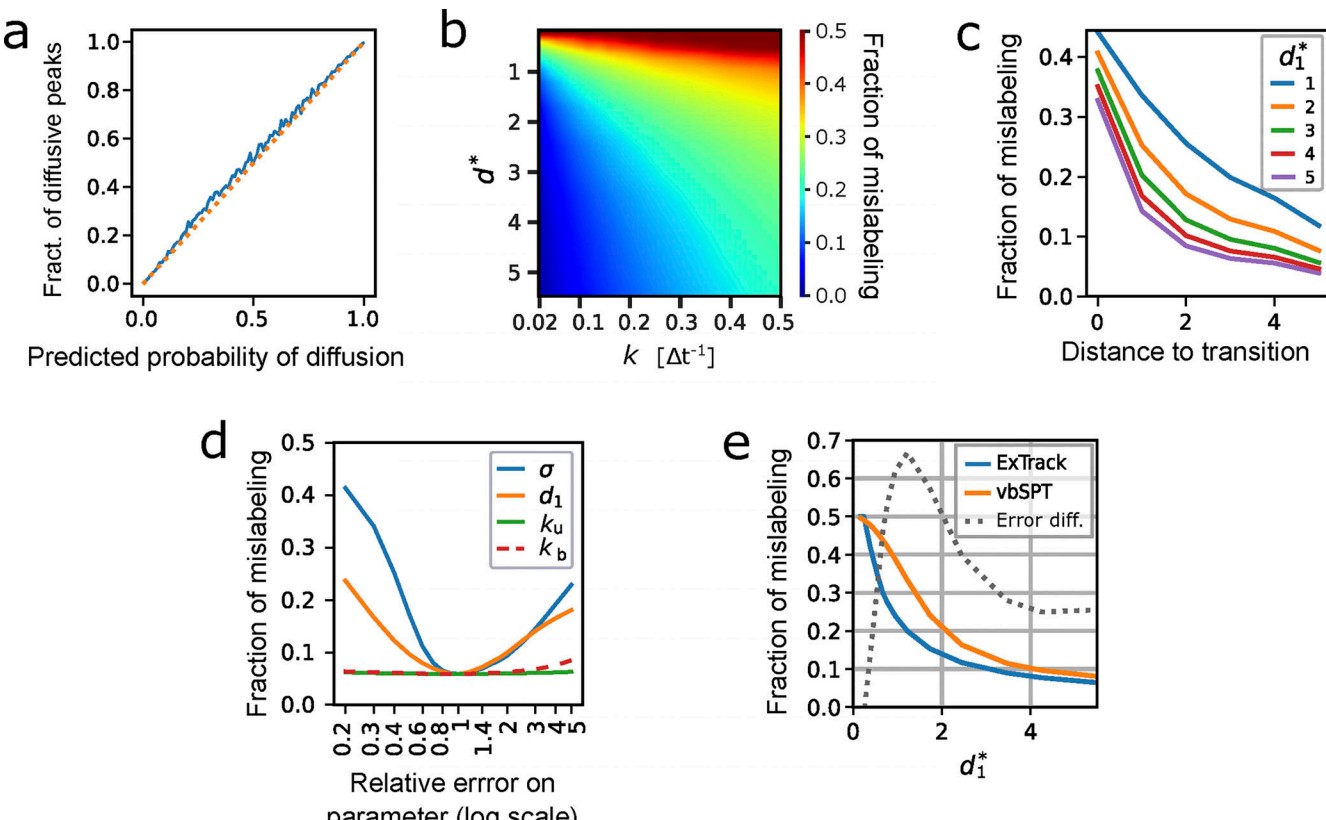

Figure S4.  **Capacity of the annotation module.** Assessment of the annotation module accuracy by comparing state estimations (either probabilistic or categorical) with ground truth from simulated tracks. If not stated otherwise, $d_0^* = 0$, $d_1^* = 5$, and $k_u = k_b = 0.1\,\Delta t^{-1}$. **(a)** Fraction of time points actually in diffusive state depending on the probability to be diffusive estimated by ExTrack. More specifically, time points are binned according to their probability to be diffusive (x axis) and for each bin we computed the fraction actually in diffusive state (y axis). Binning of 0.01. **(b–d)** Categorical state predictions are obtained by picking the state with highest probability for each time point. The fraction of mislabeled time points can thus be computed by comparison to the known true states of the simulated tracks. **(b)** Heatmap of the fractions of mislabeled time points depending on $d_1^*$ and $k$ (10,000 tracks of 10 time points). **(c)** Fraction of mislabeled time points depending on (temporal) distance to transition time points. **(d)** Fractions of mislabeled time points using correct parameters except for one of them specified in the legend. x-axis: Relative error of the varied parameter compared to the true value underlying the simulated tracks. For this particular simulation, we used 10,000 tracks of 20 time points. Window length of 10. **(e)** Fraction of mislabeled time points depending on $d_1^*$ for ExTrack and vbSPT. The grey dotted curve (annotated as Error dif.) is the relative difference of the fraction of mislabeled time points between vbSPT and ExTrack $(error_{vbSPT} - error_{ExTrack})/error_{ExTrack}$ (10,000 tracks of 10 time points).

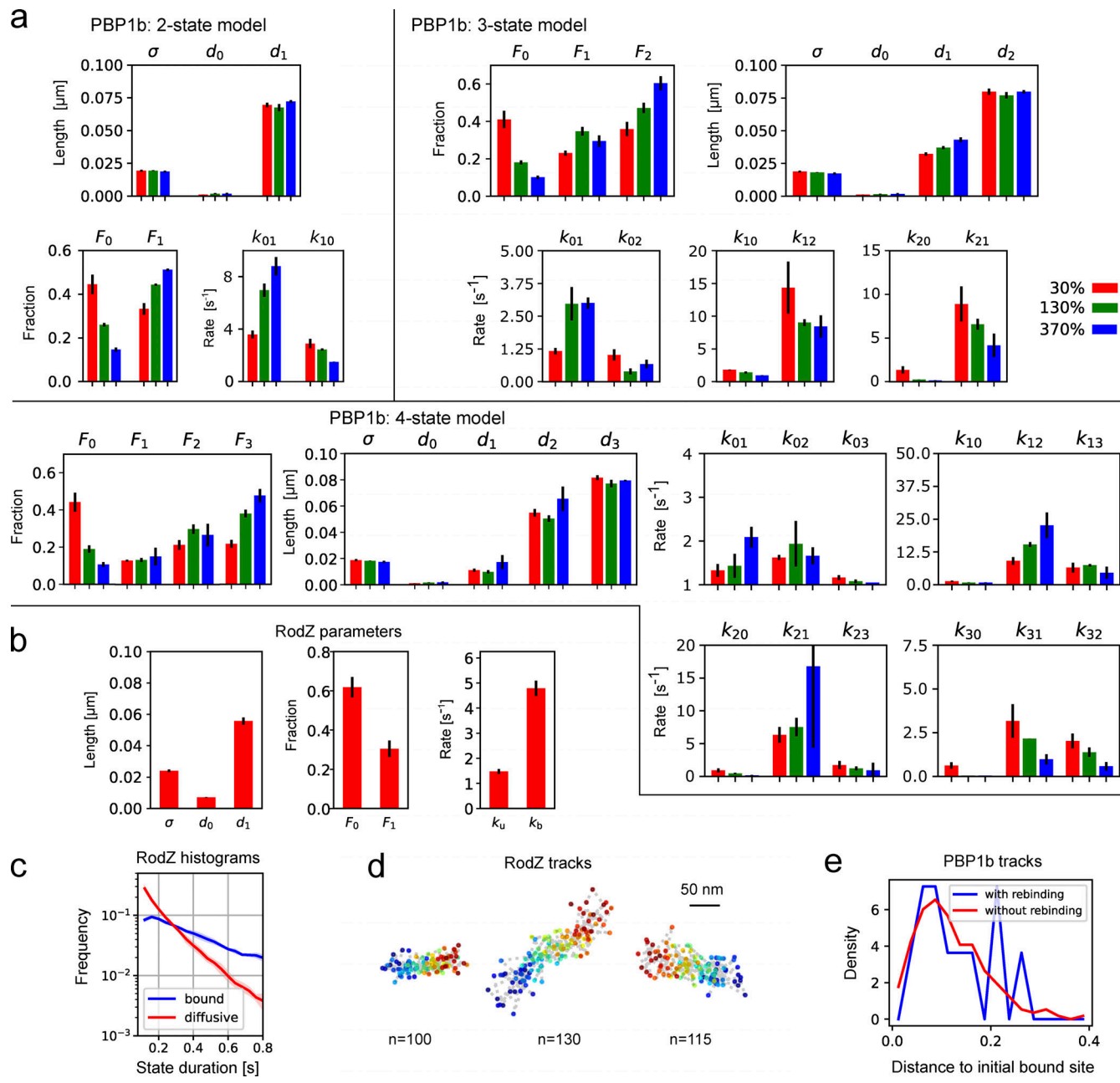

Figure S5. **Complementary results for GFP-PBP1b and GFP-RodZ tracks. (a)** Results from parameter fits to experimental tracks of GFP-PBP1b of at least three time points (and considering not more than the first 50 time points) assuming two, three, or four states in ExTrack (three replicates per condition, each replicate has at least 17,000 tracks of average lifetime from 6.3 to 7.5 positions). ExTrack settings: Window length = 4 for two/three states and 5 for four states, no sub-steps. State fractions obtained from rates. **(b–d)** ExTrack analysis of GFP-RodZ data (same as in Fig. 4, f–h). **(b)** Parameters found by ExTrack for GFP-RodZ tracks (three replicates, each replicate has at least 25,000 tracks of average lifetime from 6.3 to 7.4 positions), considering all tracks with at least three time points and restricting analysis to the 50 first time points. **(c)** Histogram of the time spent in bound or diffusive states, among tracks of at least 21 time points using parameters from b. **(d)** Examples of long bound RodZ tracks in linear motion. **(e)** PBP1b rebinding. Histograms of the distances in between initially bound position and after four diffusive steps of PBP1b tracks that are subsequently either rebinding (blue) or not rebinding (red; see Materials and methods). Histograms show no noticeable difference. Error bars and shaded regions: SDs between replicates.

Video 1. **msfGFP-PBP1b single-particle tracking.** msfGFP-PBP1b particles acquired with TIRF illumination, acquisition at a frame rate of 40 fps (displayed at 10 fps), pixel size = 130 nm. Overlay of the tracks detected with TrackMate and labeled according to two-state parameter fitting and state probability from ExTrack (jet scale: blue = bound, red = diffusive).

Video 2.   **GFP-RodZ single-particle tracking.** GFP-RodZ fusion proteins tracked with TIRF illumination, acquisition at a frame rate of 25 fps (displayed at 10 fps), pixel size = 130 nm. Overlay of the tracks detected with TrackMate and labeled according to two-state parameter fitting and state probability from ExTrack (jet scale: blue = bound, red = diffusive).

**Provided online are two tables. Table S1 shows using ExTrack to fit parameters of a three-state model to different simulations of three-state data with qualitatively different types of transitions. Table S2 shows two-state and three-state fits of tracks from simulated particles either in immobile state (state 0) or in one of five diffusive states (states 1–5).**

