## [Peer Review File · The Journal of Cell Biology]

ExTrack characterizes transition kinetics and diffusion in noisy single-particle tracks

François Simon, Jean-Yves Tinevez, and Sven van Teeffelen

Corresponding Author(s): Sven van Teeffelen, Université de Montréal

Review Timeline:

Submission Date:	2022-08-14
Editorial Decision:	2022-10-26
Revision Received:	2022-12-01
Editorial Decision:	2023-01-11
Revision Received:	2023-01-23

Monitoring Editor: Joerg Bewersdorf

Scientific Editor: Dan Simon

Transaction Report:

DOI: <https://doi.org/10.1083/jcb.202208059>

October 26, 2022

Re: JCB manuscript #202208059

Dr. Sven van Teeffelen
Université de Montréal
Microbiologie-Immunologie
Pavillon Roger-Gaudry, 2900 Edouard Montpetit Blvd
Montreal, Quebec H3T 1J4
Canada

Dear Dr. van Teeffelen,

Thank you for submitting your manuscript entitled "ExTrack characterizes transition kinetics and diffusion in noisy single-particle tracks." The manuscript was assessed by expert reviewers, whose comments are appended to this letter. We invite you to submit a revision if you can address the reviewers' key concerns, as outlined here.

You will see that the reviewers are very enthusiastic about the study and mainly request clarifications and additional explanations of the methodology. There is a request for an additional simulation of different diffusive states with their own transition rates (Reviewer2 pt#6), which we agree would be important to add.

GENERAL GUIDELINES:

Text limits: Character count for an Tools is < 40,000, not including spaces. Count includes title page, abstract, introduction, results, discussion, and acknowledgments. Count does not include materials and methods, figure legends, references, tables, or supplemental legends.

Figures: Tools may have up to 10 main text figures. Figures must be prepared according to the policies outlined in our Instructions to Authors, under Data Presentation, <https://jcb.rupress.org/site/misc/ifora.xhtml>. All figures in accepted manuscripts will be screened prior to publication.

Supplemental information: There are strict limits on the allowable amount of supplemental data. Tools may have up to 5 supplemental figures. Up to 10 supplemental videos or flash animations are allowed. A summary of all supplemental material should appear at the end of the Materials and methods section.

Please note that JCB now requires authors to submit Source Data used to generate figures containing gels and Western blots with all revised manuscripts. This Source Data consists of fully uncropped and unprocessed images for each gel/blot displayed in the main and supplemental figures. If the revised manuscript will contain cropped gel and/or blot images, please be sure to provide one Source Data file for each figure that contains gels and/or blots along with your revised manuscript files. File names for Source Data figures should be alphanumeric without any spaces or special characters (i.e., SourceDataF#, where F# refers to the associated main figure number or SourceDataFS# for those associated with Supplementary figures). The lanes of the gels/blots should be labeled as they are in the associated figure, the place where cropping was applied should be marked (with a box), and molecular weight/size standards should be labeled wherever possible. Source Data files will be made available to reviewers during evaluation of revised manuscripts and, if your paper is eventually published in JCB, the files will be directly linked to specific figures in the published article.

The typical timeframe for revisions is three to four months. While most universities and institutes have reopened labs and allowed researchers to begin working at nearly pre-pandemic levels, we at JCB realize that the lingering effects of the COVID-19 pandemic may still be impacting some aspects of your work, including the acquisition of equipment and reagents. Therefore, if you anticipate any difficulties in meeting this aforementioned revision time limit, please contact us and we can work with you to

find an appropriate time frame for resubmission. Please note that papers are generally considered through only one revision cycle, so any revised manuscript will likely be either accepted or rejected.

Thank you for this interesting contribution to Journal of Cell Biology. You can contact us at the journal office with any questions, cellbio@rockefeller.edu or call (212) 327-8588.

Sincerely,

Joerg Bewersdorf, PhD
Monitoring Editor
Journal of Cell Biology

Dan Simon, PhD
Scientific Editor
Journal of Cell Biology

Reviewer #1 (Comments to the Authors (Required)):

The manuscript by Simon et al introduced software to analyze diffusive behavior of single molecules. It showed many advantages over previous algorithms including extracting model parameters at every time point, revealing distribution of state durations, refining the positions of bound molecules, etc. The open-source code of the software makes it widely accessible and extendable for the future optimization. The manuscript is well written and organized. However, I would like to get answers for the following concerns before making final decision:

1. In the fitting module, the authors tried to extract parameters $\theta=(\sigma, D_i, F_i, k_{i,j})$ for each position using track of positions. Apparently, the number of the fitted parameters are much more than the input observed positions. Can the authors explain how could these few input data points generate so many fitted parameters?
2. The sliding window is still not at all clear to me. Smaller data points restricted by the window will reduce the complexity of the analyzing process. However, smaller data points also mean less accurate results. Furthermore, the choice of the window size is a bit arbitrary. Why a window length of 7 is used for 2-states models, and even less window length is used for 3 or more states models? How can such short windows length describe a complex model with many different states? How to choose the number of the states for a track data points?
3. In the abstract, the authors claimed that ExTrack greatly increases the regime of computationally analyzable noisy single-particle tracks. I did not find evidence to support this in the main text. Furthermore, single particle tracking techniques are normally limited to the study of relatively slow diffusion behavior since it is very hard to detect fast diffusing single molecules. It is better that the authors can comment on this and show how much the current methods could improve over the previous methods.
4. The molecule localization methods is missing in the manuscript. Since different molecule localization methods could affect the localization accuracy, it is better to mention it in the methods section.

Reviewer #2 (Comments to the Authors (Required)):

In this manuscript, Simon et al. presented a method by incorporating prior knowledge from observed position sequences into the likelihood function to estimate the parameters in Markovian based diffusion models using maximum likelihood estimate approach. In addition, they utilize a window to speed up fitting, provide different modules to refine position, and deal with non-Markov process. Through simulation, the author claimed their method has lower mean absolute error than other MLE-based methods: vbSPT and anaDDA. They also demonstrate their method's robustness under different sources of bias, localization uncertainty, and inaccurate prior knowledge. The manuscript is well-organized, and their application of estimating motion states of RodZ in cells is quite interesting. It is a remarkable job indeed from the authors in terms of technological description and useful information provided on both methods and applications. However, I do have following concerns to be addressed before my recommendation for publication of this manuscript.

Major Comments:

1. Heatmaps in Figure 1f-g show that Extrack has the smallest mean absolute error compared to vbSPT and anaDDA. However, it is unclear whether those two methods are working at their optimal condition. For example, in the anaDDA paper, they refined a set of start parameter for the optimization program for specific step numbers, and anaDDA work up to 8 steps. Could authors show in conditions of previous anaDDA or vbSPT papers that ExTrack also perform equally or in superior compared to the

existing methods? Moreover, could author elaborate on whether the simulated traces in this paper with specific parameters are suitable to apply to other methods?

2. Since ExTrack is designed to fit a larger range of parameters than other methods, could the author demonstrate the robustness of their fitting method, at what range the fitting result could converge well, and how sensitive the fitting results against the start setting (initial guess).

3. Figure 2, could the author elaborate more about how ExTrack is able to deal with physical confinement induced bias? As shown in the figure, ExTrack works well for the confining dimension two to four times larger than diffusion length. In the case, a short track, in most cases, will be very similar to a free space.

4. Figure 3a, the author only demonstrates 2 cases for the ExTrack annotate modules. Could author provide an overall performance on a statistically significant number of tracks? This is important since the isolated demonstration can hardly provide any demonstration towards general behavior of the algorithms in this situation.

6. In Supplementary Figure 3, simulating 5 diffusive states with the same transition rates is not very realistic. Could author test their method with different diffusive states with their own transition rates? Would ExTrack still work when the diffusive states have different rates?

7. The histogram module is claimed to be able to capture non-Markovian motion types. However, the author only stated the potential without explanation. Would noise also affects the shape of the histogram, and how can one quantitatively judge the motion types (Markovian or non-Markovian) in the presence of noise?

Minor Comments:

1. Line 237, Incorrect figure reference: (Fig. S??b)

2. Supplementary Figure 2b, Missing label of the gray curve

3. Supplementary Figure 2d, Missing bias curve for vbSPT

Answer to the referees

Reviewer #1 (Comments to the Authors (Required)):

The manuscript by Simon et al introduced software to analyze diffusive behavior of single molecules. It showed many advantages over previous algorithms including extracting model parameters at every time point, revealing distribution of state durations, refining the positions of bound molecules, etc. The open-source code of the software makes it widely accessible and extendable for the future optimization. The manuscript is well written and organized. However, I would like to get answers for the following concerns before making final decision:

We thank the referee for his/her appreciation of our work.

1. In the fitting module, the authors tried to extract parameters $\theta=(\sigma, D_i, F_i, k_{(i,j)})$ for each position using track of positions. Apparently, the number of the fitted parameters are much more than the input observed positions. Can the authors explain how could these few input data points generate so many fitted parameters?

$\theta=(\sigma, D_i, F_i, k_{(i,j)})$ is a set of model parameters fit to the whole data set and not to each peak. This is described in the beginning of the Results section, where we wrote: '*A fitting module fits a multi-state Markovian diffusion model to a data set of noisy single-molecule tracks. This module infers global model parameters including localization error, [...]*'. For example, in Fig. 1d, we obtained the 6 parameters of a 2-state model fit to a dataset of 10.000 tracks of 10 positions each. The Annotation Module then calculates the probability to reside in any particular state for each position. Note, that these single-time-point state probabilities are not fit parameters, but are calculated analytically based on the global model parameters.

2. The sliding window is still not at all clear to me. Smaller data points restricted by the window will reduce the complexity of the analyzing process. However, smaller data points also mean less accurate results. Furthermore, the choice of the window size is a bit arbitrary. Why a window length of 7 is used for 2-states models, and even less window length is used for 3 or more states models? How can such short windows length describe a complex model with many different states? How to choose the number of the states for a track data points?

We agree with the referee that the window method is not easy to understand. Contrary to the interpretation of the referee, the window approximation considers all data points of a track (and not only a small set of data points). As explained in detail in the Supplementary Methods, the window method approximates the calculation of the probability of observing a track by effectively averaging over different state combinations. Those averages are only performed outside of the given window size, while different state combinations inside the window are considered explicitly. Thus, all sequences of states are considered (inside the window we consider all sequences explicitly, while outside we average over different sequences).

We now write in the manuscript text: *More specifically, when recursively computing the probability density of observing a track, we need to consider all possible combinations of states up to the current time point of the integration. To save computational time, the window method*

averages over different sequences of states outside of the sliding window, while explicitly considering different combinations of states within the window (see Methods, section 1.D). Choosing the window length is a compromise between accuracy and speed.

Regarding the choice of the window size: The window allows a compromise between speed and accuracy. Shorter windows result in faster computation but less accurate approximations. For the 2-states models we could afford a higher window length, while this became computationally more demanding for the 3-state model. However, we now demonstrate in a new figure panel (Fig. S3c) that a shorter window also leads to good parameter estimates down to $d^* > 1.7$ (while a window size of 10 lead to good parameter estimates down to $d^* > 0.9$). In summary, we propose to use a window length that is as high as computationally feasible. Having said this, if d^* is known to be larger than about 2, a window length of 3 is sufficient.

To demonstrate the ability of ExTrack to fit data reliably with a lower window size, we introduced a new panel (Fig. S1c) that demonstrates how the fitting module works on 2-state data with a shorter window of 3 steps:

We now write in the main manuscript: *However, the method predicts parameters almost equally reliably without sub-steps (Fig.~S1b), or with a smaller window size of 3 (Fig.~S1c) while achieving improved computation time (Fig. S1d).*

We also added two sentences to the first part of the Results section: *We suggest using a window length of 3 to 7 depending on the expected diffusion lengths. We will show later that a window length of 3 can be sufficiently large if the ratio between diffusion length and localization precision is greater than about 2.*

3. In the abstract, the authors claimed that ExTrack greatly increases the regime of computationally analyzable noisy single-particle tracks. I did not find evidence to support this in the main text. Furthermore, single particle tracking techniques are normally limited to the study

of relatively slow diffusion behavior since it is very hard to detect fast diffusing single molecules. It is better that the authors can comment on this and show how much the current methods could improve over the previous methods.

In Figure 1f-g, we compare ExTrack to two previous methods, vbSPT and anaDDA, that fit multi-state diffusion models to SPT datasets of a broad range of parameters. This comparison shows that ExTrack allows correct parameter estimates for a wider range of dimensionless diffusion lengths ($d^* = d/\sigma$) and transition rates than both of the previous methods. More specifically, Fig. 1f-g shows that our method works for values of $d^* > 0.9$ while the previous methods are restricted to $d^* > 2.5$ (note that the 2.8-fold decrease of d^* corresponds to a 7.7-fold change of D). ExTrack also allows good estimates for a wide range of transition rates, as observed in anaDDA but unlike vbSPT. Finally, ExTrack is much more robust w.r.t. deviations from model assumptions than anaDDA (Figure 2 and Figure S3c).

4. The molecule localization methods is missing in the manuscript. Since different molecule localization methods could affect the localization accuracy, it is better to mention it in the methods section.

We thank the referee for catching this mistake. We added the following paragraph: *'Tracks were built from movies using TrackMate with LoG peak detection (estimated blob diameter of 0.5 μm , quality threshold of 15, with subpixel localization). Then, peaks were linked into tracks using the 'Simple LAP Tracker' (max distances of 0.4 μm , and 'gap' of 2).'*

Reviewer #2 (Comments to the Authors (Required)):

In this manuscript, Simon et al. presented a method by incorporating prior knowledge from observed position sequences into the likelihood function to estimate the parameters in Markovian based diffusion models using maximum likelihood estimate approach. In addition, they utilize a window to speed up fitting, provide different modules to refine position, and deal with non-Markov process. Through simulation, the author claimed their method has lower mean absolute error than other MLE-based methods: vbSPT and anaDDA. They also demonstrate their method's robustness under different sources of bias, localization uncertainty, and inaccurate prior knowledge. The manuscript is well-organized, and their application of estimating motion states of RodZ in cells is quite interesting. It is a remarkable job indeed from the authors in terms of technological description and useful information provided on both methods and applications. However, I do have following concerns to be addressed before my recommendation for publication of this manuscript.

We thank the referee for his/her careful assessment and appreciation of our work.

Major Comments:

1. Heatmaps in Figure 1f-g show that Extrack has the smallest mean absolute error compared to vbSPT and anaDDA. However, it is unclear whether those two methods are working at their optimal condition. For example, in the anaDDA paper, they refined a set of start parameter for

the optimization program for specific step numbers, and anaDDA work up to 8 steps. Could authors show in conditions of previous anaDDA or vbSPT papers that ExTrack also perform equally or in superior compared to the existing methods? Moreover, could author elaborate on whether the simulated traces in this paper with specific parameters are suitable to apply to other methods?

In the article presenting anaDDA, the authors consider up to 50.000 tracks with an average track length of 4 time points and a high value of $d^* = 9.4$. In the same conditions, and assuming a transition rate of 0.1/step, we found ExTrack to result in less than 2% error on parameters for ExTrack without substeps and less than 1% error with two substeps. This is at least as good as anaDDA, which shows parameter errors of around 3% according to their Figure 2. vbSPT also uses an exponential track-length distribution with a very similar average track length of 5 steps and with a variable number of tracks. They only simulate a 3-state model with 3 diffusive states ($d^*=1.9, 3.9, 6.7$) and different transition rates for different pairs of states. For these conditions, vbSPT shows an error of about 20% in multiple of the rates, even for the largest number of tracks (500.000). For similar parameters ($d^*=0,2,5$) with much less tracks (10.000 tracks of 10 steps each, which is about 6-fold less than the number of 10-step-long tracks in the vbSPT dataset), ExTrack shows lower errors of maximally 5%.

In our manuscript, we focused on measuring the capability of ExTrack to estimate parameters from particles with an immobile and one or multiple diffusive states. We chose track lengths of 10 time points to have standard data sets knowing that the track length distribution has little impact in the cases of tracks that cannot leave the field of view (as also simulated for vbSPT and anaDDA). Overall, we think that our heatmaps demonstrate the capability of ExTrack to be as good or better than the previous methods for a wide range of parameters. However, we do not aim to demonstrate that ExTrack is much superior than anaDDA and vbSPT for high d^* . Instead, we focus on demonstrating that ExTrack works in a larger parameter range: Contrary to ExTrack, tracks of low d^* are not suitable for anaDDA or vbSPT, and tracks of high transition rates are generally not suitable for vbSPT, as we already say in the manuscript.

2. Since Extrack is designed to fit a larger range of parameters than other methods, could the author demonstrate the robustness of their fitting method, at what range the fitting result could converge well, and how sensitive the fitting results against the start setting (initial guess).

For the full range of parameters tested in the heatmap of Fig. 1f, we use a single set of initial parameters, which are now indicated in the figure caption. Despite the same initial condition the method converges well. In the following we varied one of those initial parameters, $D1$, for a single simulation of tracks with ($D1=0.04 \text{ um}^2/\text{s}$, $k=0.1/\text{step}$, $\sigma=0.02 \text{ um}$, $dt = 0.02\text{s}$, resulting in $d^*=2$), and found hardly any variations in the predicted parameter values:

We now write in the manuscript: *These estimates are robust with respect to the initial parameters (for example if varying the initial d_1^* between 0.3 and 10).*

3. Figure 2, could the author elaborate more about how ExTrack is able to deal with physical confinement induced bias? As shown in the figure, ExTrack works well for the confining dimension two to four times larger than diffusion length. In the case, a short track, in most cases, will be very similar to a free space.

In its current form, ExTrack does not consider confinement explicitly. This has practical reasons: First, considering confinement would require the user to indicate the dimensions of the confining space. Second, and more importantly, it would render the distribution of physical displacements non-Gaussian, which, in turn, would impede the analytical calculation of the probability of observing a track (Eq. 6). As a consequence, the estimated diffusion length is underestimated if the confining box size is smaller than about $4 d^*$. Interestingly though, ExTrack is still able to correctly predict transition rates even for box sizes as small as $2 d^*$.

We added the following sentence to the Results section: *'Interestingly, while the diffusion length is underestimated for confining dimensions smaller than about $4 d^*$, the transition rates are predicted reliably even if the confining box size is as small as $2 d^*$.'*

4. Figure 3a, the author only demonstrates 2 cases for the ExTrack annotate modules. Could author provide an overall performance on a statistically significant number of tracks? This is important since the isolated demonstration can hardly provide any demonstration towards general behavior of the algorithms in this situation.

We assessed the performance of the annotation module in two different ways: 1) We considered its probabilistic annotation. 2) We considered a categorical annotation based on a threshold probability of 50% (e.g., if the probability to be diffusive is predicted to be >50% the time point is annotated as diffusive). To demonstrate the reliable probabilistic annotation of ExTrack, we now introduced a new figure panel Fig. 3c, which shows the distribution of the probability to be diffusive at the time points where the particles are actually diffusive according to the ground truth (simulation):

Distribution of the probability to be diffusive at the time points where the particles are actually diffusive. Similar simulations than in a for different d^ .*

This plot demonstrates that ExTrack annotates diffusive molecules as very likely diffusive if $d^* \geq 2$, while the distribution of probabilities is very broad for lower values of d^* .

Additionally, we already show in Fig. S4c (previously S4b) that among all molecules predicted to reside in the diffusive state with probability p , the fraction of molecules actually diffusive also equals p .

Regarding the categorical annotation, we show in Fig. S4b (previously S4d) the error of annotation for a large range of diffusion lengths and transition parameters. Supplementary Figure 4e shows how ExTrack performs compared to vbSPT.

For more clarity the paragraph on state annotation was reworked. We now write:

Next, we tested the performance of the single-molecule probabilistic annotation module of ExTrack, which is based on global model parameters and annotates state probabilities for every time point (31). Fig. 3a-b shows tracks from the simulation of a two-state model with an immobile and a slowly diffusive state ($d^ = 2$). Despite the small value of d^* , motion states are reliably estimated in this example. Then, we estimated the predictive power of the probabilistic annotation module by applying it to large sets of tracks with different values of d^* (0.5, 1 or 2). As expected, higher d^* values result in higher confidence in predicting states (Fig. 3c). To demonstrate the accuracy of the probabilistic annotation, we confirmed that among all molecules predicted to reside in the diffusive state with probability p , the fraction of molecules actually diffusive also equals p (Fig. S4a). Previous methods often classify molecules*

categorically into the most likely state (3, 32, 47). We used this approach to measure the performance of ExTrack and compare it to previous methods: First, we estimated the accuracy of categorical state annotations depending on the diffusion length and the transition rates (Fig. S4b). Increasing transition rates lead to worse estimates, because states are easier to estimate at time points that are distant from transitions (Fig. S4c). Second, we found ExTrack annotations to be robust with respect to wrongly chosen global parameters (Fig. S4d). Finally, we found that the ExTrack state annotation module performs better than vbSPT (Fig. S4e).

Supplementary Figure 4:

6. In Supplementary Figure 3, simulating 5 diffusive states with the same transition rates is not very realistic. Could author test their method with different diffusive states with their own transition rates? Would ExTrack still work when the diffusive states have different rates?

This is a good point. We changed the simulation by picking transition rates from a broad range between 0.01 and 0.2. Figure S3b is now updated. Exact rates are $k_{01} = 0.03$, $k_{02} = 0.02$, $k_{03} = 0.06$, $k_{04} = 0.04$, $k_{05} = 0.05$, $k_{10} = 0.04$, $k_{12} = 0.05$, $k_{13} = 0.1$, $k_{14} = 0.07$, $k_{15} = 0.04$, $k_{20} = 0.19$, $k_{21} = 0.01$, $k_{23} = 0.04$, $k_{24} = 0.06$, $k_{25} = 0.1$, $k_{30} = 0.02$, $k_{31} = 0.03$, $k_{32} = 0.05$, $k_{34} = 0.05$, $k_{35} = 0.1$, $k_{40} = 0.1$, $k_{41} = 0.04$, $k_{42} = 0.04$, $k_{43} = 0.01$, $k_{45} = 0.05$, $k_{50} = 0.15$, $k_{51} = 0.05$, $k_{52} = 0.01$, $k_{53} = 0.02$, $k_{54} = 0.05$ (in Δt^{-1}).

The parameter estimate works reliably:

7. The histogram module is claimed to be able to capture non-Markovian motion types. However, the author only stated the potential without explanation. Would noise also affect the shape of the histogram, and how can one quantitatively judge the motion types (Markovian or non-Markovian) in the presence of noise?

We are not sure which type of noise the referee refers to. Here, we address how to deal with two types of noise:

First, the histogram module takes into account the noisy nature of tracks (localization error and Brownian motion) by weighing the contributions of segments according to the certainty of segment annotation. We now emphasize this point in the manuscript by writing: *Instead of considering only the most likely set of states, ExTrack considers a large number of potential state vectors and weighs them with their corresponding probabilities.*

Second, exponential behavior can only be confirmed or ruled out if histograms themselves are not too noisy, meaning if there are enough long segments to populate the state-duration histogram, which can be confirmed by observing overlapping curves from replicates.

Minor Comments:

1. Line 237, Incorrect figure reference: (Fig. S??b)

We corrected the figure reference Fig. S3b.

2. Supplementary Figure 2b, Missing label of the gray curve

We corrected this mistake and now introduced the label: 'b: Results from UncertainSPT for different diffusion lengths. Gray curve: ideal model.'

3. Supplementary Figure 2d, Missing bias curve for vbSPT

We now added this curve.

January 11, 2023

RE: JCB Manuscript #202208059R

Dr. Sven van Teeffelen
Université de Montréal
Microbiologie-Immunologie
Pavillon Roger-Gaudry, 2900 Edouard Montpetit Blvd
Montreal, Quebec H3T 1J4
Canada

Dear Dr. van Teeffelen,

Thank you for submitting your revised manuscript entitled "ExTrack characterizes transition kinetics and diffusion in noisy single-particle tracks." We would be happy to publish your paper in JCB pending final revisions necessary to meet our formatting guidelines (see details below).

A. MANUSCRIPT ORGANIZATION AND FORMATTING:

1) Text limits: Character count for Tools is < 40,000, not including spaces. Count includes title page, abstract, introduction, results, discussion, and acknowledgments. Count does not include materials and methods, figure legends, references, tables, or supplemental legends.

2) Figure formatting: Tools may have up to 10 main text figures. Scale bars must be present on all microscopy images, including inset magnifications. Please avoid pairing red and green for images and graphs to ensure legibility for color-blind readers. If red and green are paired for images, please ensure that the particular red and green hues used in micrographs are distinctive with any of the colorblind types. If not, please modify colors accordingly or provide separate images of the individual channels.

3) Statistical analysis: Error bars on graphic representations of numerical data must be clearly described in the figure legend. The number of independent data points (n) represented in a graph must be indicated in the legend. Please, indicate whether 'n' refers to technical or biological replicates (i.e. number of analyzed cells, samples or animals, number of independent experiments). If independent experiments with multiple biological replicates have been performed, we recommend using distribution-reproducibility SuperPlots (please see Lord et al., JCB 2020) to better display the distribution of the entire dataset, and report statistics (such as means, error bars, and P values) that address the reproducibility of the findings.

Statistical methods should be explained in full in the materials and methods. For figures presenting pooled data the statistical measure should be defined in the figure legends. Please also be sure to indicate the statistical tests used in each of your experiments (both in the figure legend itself and in a separate methods section) as well as the parameters of the test (for example, if you ran a t-test, please indicate if it was one- or two-sided, etc.). Also, if you used parametric tests, please indicate if the data distribution was tested for normality (and if so, how). If not, you must state something to the effect that "Data distribution was assumed to be normal but this was not formally tested."

4) Materials and methods: Should be comprehensive and not simply reference a previous publication for details on how an experiment was performed. Please provide full descriptions (at least in brief) in the text for readers who may not have access to referenced manuscripts. The text should not refer to methods "...as previously described."

5) For all cell lines, vectors, constructs/cDNAs, etc. - all genetic material: please include database / vendor ID (e.g., Addgene, ATCC, etc.) or if unavailable, please briefly describe their basic genetic features, even if described in other published work or gifted to you by other investigators (and provide references where appropriate). Please be sure to provide the sequences for all of your oligos: primers, si/shRNA, RNAi, gRNAs, etc. in the materials and methods. You must also indicate in the methods the source, species, and catalog numbers/vendor identifiers (where appropriate) for all of your antibodies, including secondary. If antibodies are not commercial, please add a reference citation if possible.

6) Microscope image acquisition: The following information must be provided about the acquisition and processing of images:
a. Make and model of microscope
b. Type, magnification, and numerical aperture of the objective lenses

- c. Temperature
- d. Imaging medium
- e. Fluorochromes
- f. Camera make and model
- g. Acquisition software
- h. Any software used for image processing subsequent to data acquisition. Please include details and types of operations involved (e.g., type of deconvolution, 3D reconstitutions, surface or volume rendering, gamma adjustments, etc.).

7) References: There is no limit to the number of references cited in a manuscript. References should be cited parenthetically in the text by author and year of publication. Abbreviate the names of journals according to PubMed.

8) Supplemental materials: Tools papers may have up to 5 supplemental figures and 10 videos. Please also note that tables, like figures, should be provided as individual, editable files. A summary of all supplemental material should appear at the end of the Materials and methods section. Please include one brief sentence per item.

9) Video legends: Should describe what is being shown, the cell type or tissue being viewed (including relevant cell treatments, concentration and duration, or transfection), the imaging method (e.g., time-lapse epifluorescence microscopy), what each color represents, how often frames were collected, the frames/second display rate, and the number of any figure that has related video stills or images.

10) eTOC summary: A ~40-50 word summary that describes the context and significance of the findings for a general readership should be included on the title page. The statement should be written in the present tense and refer to the work in the third person. It should begin with "First author name(s) et al..." to match our preferred style.

11) Conflict of interest statement: JCB requires inclusion of a statement in the acknowledgements regarding competing financial interests. If no competing financial interests exist, please include the following statement: "The authors declare no competing financial interests." If competing interests are declared, please follow your statement of these competing interests with the following statement: "The authors declare no further competing financial interests."

12) A separate author contribution section is required following the Acknowledgments in all research manuscripts. All authors should be mentioned and designated by their first and middle initials and full surnames. We encourage use of the CRediT nomenclature (<https://casrai.org/credit/>).

13) ORCID IDs: ORCID IDs are unique identifiers allowing researchers to create a record of their various scholarly contributions in a single place. At resubmission of your final files, please consider providing an ORCID ID for as many contributing authors as possible.

14) Materials and data sharing:

As a condition of publication, authors must make protocols and unique materials (including, but not limited to, cloned DNAs; antibodies; bacterial, animal, or plant cells; and viruses) described in our published articles freely available upon request by researchers, who may use them in their own laboratory only. All materials must be made available on request and without undue delay. We strongly encourage to deposit all the cell lines/strains and reagents generated in this study in public repositories.

All datasets included in the manuscript must be available from the date of online publication, and the source code for all custom computational methods, apart from commercial software programs, must be made available either in a publicly available database or as supplemental materials hosted on the journal website. Numerous resources exist for data storage and sharing (see Data Deposition: <https://rupress.org/jcb/pages/data-deposition>), and you should choose the most appropriate venue based on your data type and/or community standard. If no appropriate specific database exists, please deposit your data to an appropriate publicly available database.

B. FINAL FILES:

****It is JCB policy that if requested, original data images must be made available to the editors. Failure to provide original images upon request will result in unavoidable delays in publication. Please ensure that you have access to all original data images prior to final submission.****

****The license to publish form must be signed before your manuscript can be sent to production. A link to the electronic license to publish form will be sent to the corresponding author only. Please take a moment to check your funder requirements before choosing the appropriate license.****

Thank you for this interesting contribution, we look forward to publishing your paper in Journal of Cell Biology.

Sincerely,

Joerg Bewersdorf, PhD
Monitoring Editor
Journal of Cell Biology

Dan Simon, PhD
Scientific Editor
Journal of Cell Biology

Reviewer #1 (Comments to the Authors (Required)):

The authors have successfully addressed my concerns. Congratulations on this excellent development that will be very useful to quantify the single molecule tracks.